# Global potential for harvesting drinking water from air using solar energy

Jackson Lord[1]✉, Ashley Thomas[1], Neil Treat[1], Matthew Forkin[1], Robert Bain[2], Pierre Dulac[3], Cyrus H. Behroozi[1], Tilek Mamutov[1], Jillia Fongheiser[1], Nicole Kobilansky[1], Shane Washburn[1], Claudia Truesdell[1], Clare Lee[1] & Philipp H. Schmaelzle[1]✉

Access to safely managed drinking water (SMDW) remains a global challenge, and affects 2.2 billion people[1,2]. Solar-driven atmospheric water harvesting (AWH) devices with continuous cycling may accelerate progress by enabling decentralized extraction of water from air[3–6], but low specific yields (SY) and low daytime relative humidity (RH) have raised questions about their performance (in litres of water output per day)[7–11]. However, to our knowledge, no analysis has mapped the global potential of AWH[12] despite favourable conditions in tropical regions, where two-thirds of people without SMDW live[2]. Here we show that AWH could provide SMDW for a billion people. Our assessment—using Google Earth Engine[13]—introduces a hypothetical 1-metre-square device with a SY profile of 0.2 to 2.5 litres per kilowatt-hour (0.1 to 1.25 litres per kilowatt-hour for a 2-metre-square device) at 30% to 90% RH, respectively. Such a device could meet a target average daily drinking water requirement of 5 litres per day per person[14]. We plot the impact potential of existing devices and new sorbent classes, which suggests that these targets could be met with continued technological development, and well within thermodynamic limits. Indeed, these performance targets have been achieved experimentally in demonstrations of sorbent materials[15–17]. Our tools can inform design trade-offs for atmospheric water harvesting devices that maximize global impact, alongside ongoing efforts to meet Sustainable Development Goals (SDGs) with existing technologies.

Ensuring reliable access to safe drinking water for all remains a global challenge, and is formally recognized as an international development priority by 2030 in the United Nations framework for global development priorities, the Sustainable Development Goals 6.1[18]. Progress towards this target is measured by the WHO/UNICEF Joint Monitoring Programme (JMP) as the percentage of population using safely managed drinking water (SMDW), where 'safely managed' is defined as "an improved source located on the premises, available when needed and free of fecal and priority chemical contamination"[1,2]. Traditional routes to bring SMDW on premises to currently unserved populations are estimated to cost US$114 billion per year (from 2015), more than three times the historical financing trend[19]. Moreover, there is increasing global interest in solutions that provide safe drinking water without the environmental consequences of increasing reliance on bottled water and that do not require household-level intervention, which has limited adherence[20,21]. Atmospheric water harvesting (AWH) shows promise to accelerate decentralized access to underserved communities if a cost-effective, off-grid device can be designed and scaled[6].

Several classes of off-grid AWH designs exist or are being explored[8,12,22,23], as summarized in Table 1. AWH devices are categorized by energy source—active devices use external energy sources whereas passive devices rely solely on atmospheric conditions that allow for pre-condensed dew or fog to be harvested. Passive devices are thus limited to geographic niches where dew or fog can be systematically harvested[7,12,24]. Active, sorbent-based AWH devices extract water using primarily solar thermal energy in one of two operational modes: diurnal-mode devices extract at night (when RH is higher) and condense during the day (when solar energy is available) in a single daily cycle, requiring a large sorbent bed. By contrast, continuous-mode devices are not limited to a single daily cycle, and need only hold a small amount of water vapour in-process[3,4], drastically reducing sorbent mass and device size. This, however, requires extraction at lower RH when solar energy is available, raising questions about performance[7–11]. Cooler–condenser devices use work (typically electric energy) to actively cool air below its dew point and collect condensation and—if solar-driven—call for photovoltaic (PV) panels. Unlike solar–thermal devices, solar-driven cooler–condenser devices suffer from a steep loss in electric energy conversion. In the context of specific yield, we use kWh to denote primary solar energy prior to thermal and other losses, and $kWh_{PV}$ to denote electrical energy supplied to the device from PV panels after conversion. Unless stated otherwise, ranges of SY refer to RH between 30% and 90% at 20 °C.

Here we present an assessment of solar-driven, continuous-mode AWH (SC-AWH) using global data. AWH has much lower SY than

[1]X, The Moonshot Factory, Mountain View, CA, USA. [2]WHO/UNICEF Joint Monitoring Programme, Division of Data, Analytics, Planning and Monitoring, UNICEF, New York, NY, USA. [3]Google Inc., Mountain View, CA, USA. ✉e-mail: jacksonlord@gmail.com; phs@google.com

## Table 1 | Suitability of household-scale applications by AWH category

| | Passive AWH devices | Active AWH devices |
|---|---|---|
| **Diurnal AWH (single cycle per day)** | | |
| Device types: | Dew harvesters (near-condensed droplets) | Sorption-based |
| Energy Requirements: | None | 0 to 1 l kWh$^{-1}$ (ref. [32]) |
| Size requirements: | Low mass, but requires large catchment surface area | Mass-driven: water outputs scale proportional to sorbent mass[37] |
| Global assessment: | Niche potential[7] | Wide climate applicability but mass intensity limits economic reach[7,31] |
| **Continuous AWH (or multiple cycles per day)** | | |
| Device types: | Fog harvesters (pre-condensed droplets) | Sorption-based, cooler–condensers* |
| Energy requirements: | None | Sorption-based: 0 to 1 l kWh$^{-1}$ (ref. [32]); cooler–condenser: 2 to 4 l kWh$^{-1}$ (ref. [32])* |
| Sizing requirements: | Low mass, but requires large catchment surface area | Climate-driven and modular: scaled by available resource and solar harvesting area[4]* |
| Global assessment: | Niche potential[7] | Global potential not previously studied[12]† |

Select categorization of AWH devices with low or no energy requirements. *Promising categories for low-cost, off-grid devices at household scale. †There is a gap in the literature on global assessment, which is addressed in this study.

infrastructural water sources such as desalination[25] (approximately 200 l kWh$^{-1}$). However, SC-AWH devices sized to produce sufficient daily drinking water output for an individual or family could address both the water quality and the water access dimensions of SMDW solutions at the household level.

## Geography of the global challenge

To estimate the impact potential of SC-AWH, we first mapped the distribution of the approximately 2.2 billion people without SMDW[2]. Recent studies have used geostatistical techniques to estimate subnational inequalities of safe water and sanitation from a variety of data sources reporting metrics of facility type[26,27]. Here we use a deterministic method based exclusively on JMP data on drinking water service levels. In this study, we assume that SC-AWH is for drinking water only and does not replace water for other domestic uses such as hygiene, cooking and sanitation[14,28].

The overall percentage of the population in regions reported by the JMP at the lowest respective available regional hierarchy is shown in Fig. 1a. This seamless fabric of national and subnational survey regions gives a spatially continuous picture of the global distribution of people living without SMDW. Sub-Saharan Africa contains the highest total number of people without SMDW, in alignment with previous reports[2,29], followed by regions in South Asia and Latin America.

The regional proportions from Fig. 1a were applied as a linear weight to each pixel of the WorldPop (2017) 1 km-resolution residential population counts image (https://www.worldpop.org). This gives an estimate of the distribution of people without SMDW to a spatial resolution that more closely matches the scales at which climate variables relevant for AWH vary owing to physical geography, such as topography and land cover. The resulting weighted population distribution is shown in Fig. 1b.

## Geospatial toolset for AWH assessment

We present a geospatial tool (AWH-Geo) for assessing the global potential for notional SC-AWH devices given available climatic resources. AWH-Geo was built in Google Earth Engine[13] and is extensible across climate data. For this study, AWH-Geo uses the ERA5-Land climate reanalysis over the 10-year period 2010–2019 (inclusive). ERA5-Land was chosen for its fine resolution (9 km at hourly intervals), global coverage and ability to represent historical synoptic conditions. This period is sufficient to account for interannual variability, although decadal trends are explored in brief in Extended Data Fig. 9. For shorter computation times running their own analysis, the user can adjust the analysis period within the tool.

AWH-Geo takes as input the instantaneous rate of water output as a function of the three dominant environmental variables: (1) global horizontal irradiance from sunlight (GHI (in W m$^{-2}$)), (2) RH (%) and (3) air temperature ($T$ (°C)). Secondary climate variables could be incorporated later (for example, downwelling infrared and surface wind speed). We propose an output table with water yield values as a function of binned climate inputs GHI, RH and $T$, as a way to connect AWH device models or experimental characterizations with geospatial analyses. Water output can be entered in areal harvesting rates (in l h$^{-1}$ m$^{-2}$) for abstractions, or as the expected yield of a real device with known collection areas (in l h$^{-1}$). Across all data points of a multi-year climate image time series, AWH-Geo uses the given output table to look up yield values and aggregates water outputs for display as global maps or derived plots. Whereas previous assessments have been limited to relatively small numbers of locations with on-site meteorological data[7,30] or limited the analysis to a region[31], the approach presented here is global and spatially continuous. Figure 2 shows a conceptual workflow of AWH-Geo and adjacent processes to produce results in this study.

We first used AWH-Geo to map theoretical upper bounds of solar-driven AWH by constructing output tables from the literature as specific water yields SY (in l kWh$^{-1}$). SY is an evaluative metric for AWH sensitive to RH[32], and is the inverse of specific energy consumption (SEC), which is commonly used for other water and desalination systems. Resulting maps are overlaid with a dot-density representation of the distribution of people without SMDW for visual comparison in Fig. 3.

Recently, Kim et al. have described the fundamental thermodynamic limits for AWH[33]. This model gives the minimum thermal energy required (at a given hot-side temperature level) per unit water output of a black box AWH, corresponding to SY values between 5 and 50 l kWh$^{-1}$. Kim's thermodynamic limits are mapped in Fig. 3a. Mapping thermodynamic limits is useful to set maximum expectations for SC-AWH output globally and to assess the improvement potential that may exist between existing device performance and fundamental physical limits. Similar analytic approaches have been used to assess condenser-based devices, diurnal devices and dew collectors applied to a specific location or region[7,12,30,31]. The geographic patterns of output closely follow time-averaged humidity values generally, modified by the availability of sunlight. Notably, the results show significant water production potential throughout much of the world, particularly in the tropics.

Next, we mapped the maximum output of two basic design types. Peeters describes the maximum yield for active cooler–condensers[32], giving SYs of 1–30 l kWh$_{PV}^{-1}$ (0.2–6 l kWh$^{-1}$), plotted using AWH-Geo in Fig. 3b. For sorbent designs, metal organic frameworks (MOFs) and thermo-responsive polymer (TRP) gels[17] show the highest yields at low and high RH, respectively. Zhao et al. demonstrated exceptional performance of a TRP[15] at high RH (0.2–9.3 l kWh$^{-1}$ (converted to SY by Peeters[32])), generally outperforming MOFs (whose reported maximum[32] SYs are around 1 l kWh$^{-1}$). Global projections for Zhao's TRP are mapped in Fig. 3c.

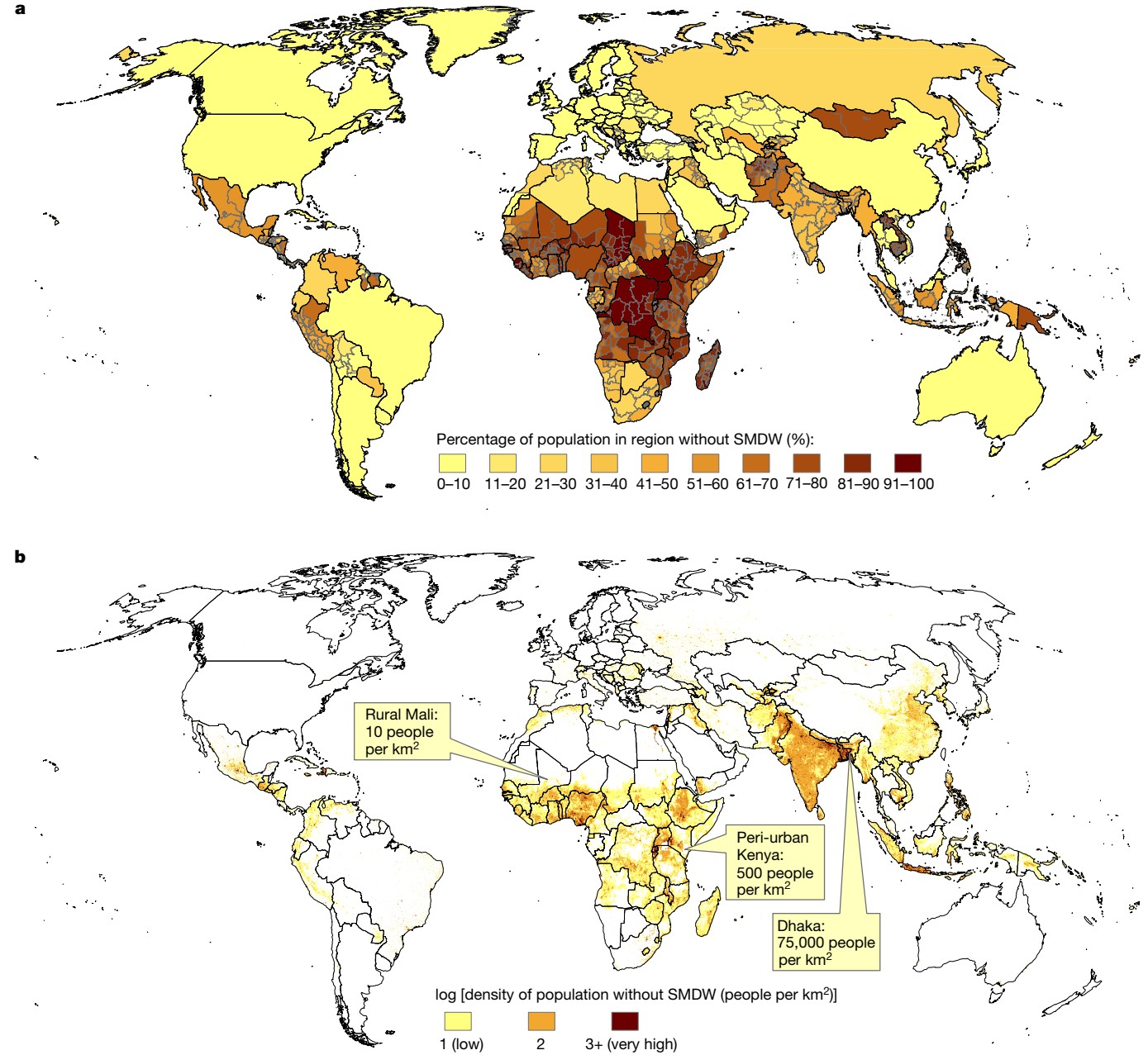

**Fig. 1 | Geographic distribution of world population without SMDW.**
**a**, Percentage share of total population in survey region living without SMDW as reported by the WHO/UNICEF JMP. **b**, Log population density of people without SMDW from WorldPop at 1 km resolution adjusted by JMP proportions at 1 km resolution. Produced in ArcGIS 10.

In addition to annual means, AWH-Geo is capable of deriving metrics useful for analysing seasonal variability of output. Optionally, AWH-Geo exports 90% availability (P90) values across a set of time windows (Methods).

## Assessing the global potential

Our coincidence analysis calculates the mean hours per day during which GHI and RH are simultaneously above parametric thresholds. Fig. 4a maps annual means for such daily coincidence hours for the given threshold pairs, interpreted as the operational hours per day (ophd) for a hypothetical device. Important transition areas between tropical and desert regions show the expected trade-off between sunlight and humidity, which generally vary inversely. Very low RH thresholds of 10% increase ophd potential by only 1–2 h from the ophd at 30% RH in arid regions in the Sahel across GHI thresholds, but ophd then falls sharply at higher RH thresholds. This indicates a diminishing return to devices operating below 30%. Coastal areas show promise for consistent 2–4 ophd worldwide above 50% RH.

Next, we summed the population without access to SMDW segmented by threshold pair using the weighted population image, grouped cumulatively by ophd at whole intervals and shown in Fig. 4b. Inflections of diminishing user potential occur between values of RH between 30 and 50%, GHI between 400 and 600 W m$^{-2}$ and ophd between 3 and 5 h. These reflect key spatio-demographic patterns along similar climatic transitions in the tropics, where the bulk of those living without SMDW live—particularly in the tropical savanna of sub-Saharan Africa and the Ganges River Valley in India. A device that could operate above these values has the theoretical potential to serve more than half the world's remaining population lacking access to SMDW.

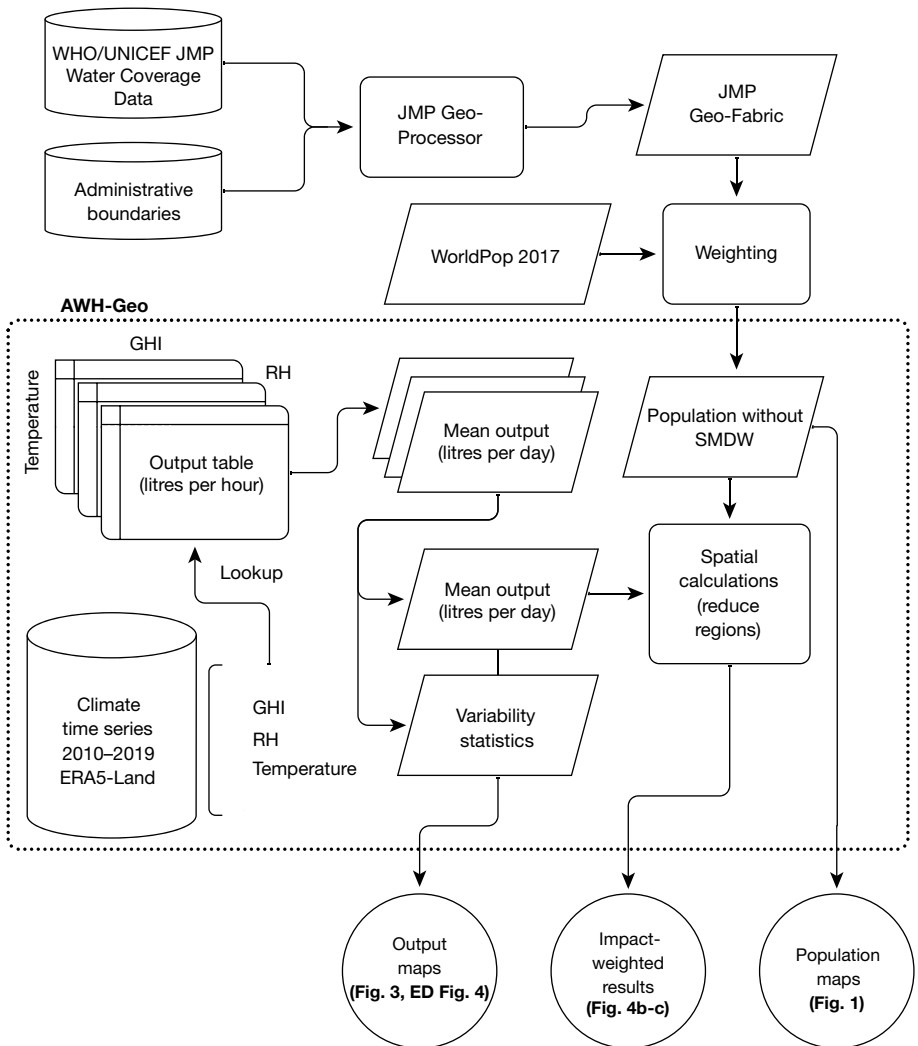

**Fig. 2 | Data processing workflow of AWH-Geo.** Cylinders indicate data stores from Google Earth Engine, the WHO/UNICEF JMP or open online content. Shown are processes (rectangles), geo-images (parallelograms) and outputs (circles).

Next we ran the SY profiles of a collection of SY curves through AWH-Geo, including commercial cooler–condenser devices evaluated by Bagheri[34] and a data sheet for the SOURCE panel, a sorbent-based device from company SOURCE, formerly known as Zero Mass Water[35] (ZMW).

Figure 4c shows resulting outputs normalized by area (in $l\,d^{-1}\,m^{-2}$)—a performance metric advocated by LaPotin et al.[11]—as a function of the population without SMDW reached. Steep gradients of the human impact of the output mirror those in the coincidence analysis. Linear SY profiles prioritize performance at low RH, but cap output even in resource-rich climates. The target curves are based on hypothetical SY values similar to those characteristic of sorbent or device profiles that reach 1 billion users at an average of $5\,l\,d^{-1}\,m^{-2}$. Comparing the two target curves demonstrates the expected trade-off between serving more users at low output (linear) and fewer users at high output (logistic).

To further explore trade-offs of the SY curve across different values of RH, we plotted SY values from materials and devices in relation to target curves for reaching 0.5–2.0 billion people without SMDW at $5\,l\,d^{-1}$, the approximate daily drinking water requirements of an individual[14] (Fig. 4d). We based the target curves on a $1\,m^2$ device unless otherwise noted, although water output and SY targets scale linearly with device area in sunlight. To demonstrate this, we plotted a version of the 1.0-billion target based on $2\,m^2$—this doubling of the device area halves the SY requirements for the target impacts. The existing devices both follow approximately linear yields across RH below the 0.5-billion impact target curves. MOFs and other sorbents show varied results[3,36], although they remain roughly linear. Zhao's exceptional yields at high RH make up for low performance at low RH (logistic profile), and show the most promise for reaching the largest user base (2.0 billion). Figure 4d compares material and device performance side-by-side to show the gap between present capabilities and theoretical limits, although real devices will be subject to losses that will prevent them from fully reaching idealized material performance or theoretical limits.

## Closing the gap

This study presents initial conclusions—developing detailed SC-AWH design criteria will require further work. A device with a $1\,m^2$ solar collection area and a SY profile of $0.2$–$2.5\,l\,kWh^{-1}$ ($0.1$–$1.25\,l\,kWh^{-1}$ for $2\,m^2$) can serve the SMDW needs of about 1 billion people, assuming continuous harvesting of 2–3 h per day of coincident sunlight of more than $600\,W\,m^{-2}$ and RH above 30%. The shape of the SY curve is critical for SC-AWH to take advantage of coincident humidity and solar energy during key periods of the day, typically during morning and evening hours. A trade-off exists between increasing yields at lower RH (around 30%) for those in climate transition zones (northern sub-Saharan Africa and western India), versus focusing on exponentially higher yields in humid regions such as Bangladesh and equatorial regions.

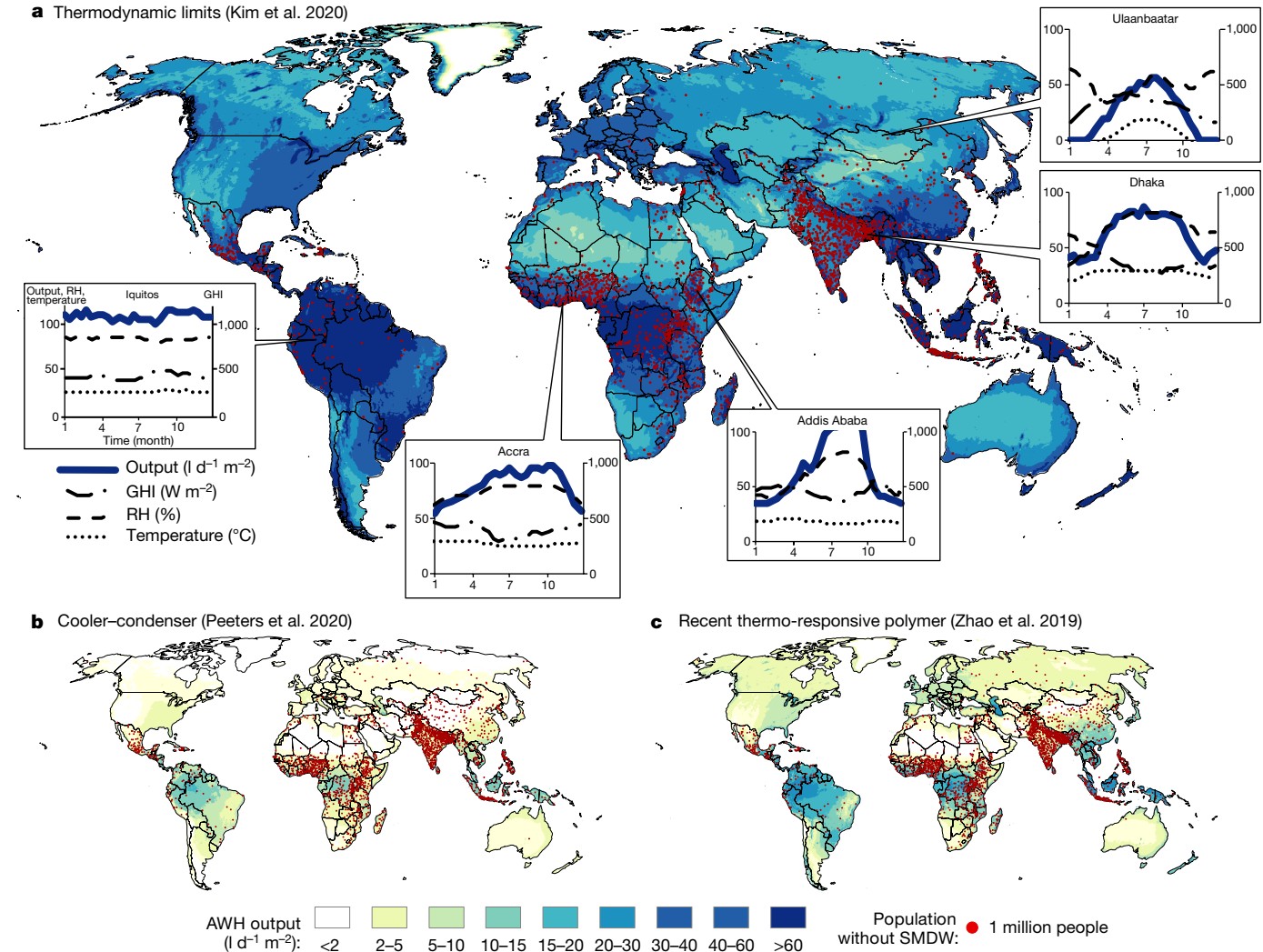

**a** Thermodynamic limits (Kim et al. 2020)

Ulaanbaatar

Dhaka

Output, RH, temperature    Iquitos    GHI

Accra

Addis Ababa

— Output (l d⁻¹ m⁻²)
-·- GHI (W m⁻²)
--- RH (%)
···· Temperature (°C)

**b** Cooler–condenser (Peeters et al. 2020)

**c** Recent thermo-responsive polymer (Zhao et al. 2019)

AWH output (l d⁻¹ m⁻²):  <2  2–5  5–10  10–15  15–20  20–30  30–40  40–60  >60

Population without SMDW:  ● 1 million people

**Fig. 3 | Upper bounds water output of solar-driven AWH in relation to global user base. a–c,** Mean daily water output of solar-driven AWH given overall thermodynamic limits of any process[33] ($T_{hot}$ = 100 °C) (**a**), cooler–condenser processes driven by PV[32] (**b**) and example of active sorbent device types (TRP gels from ref. [15]) (**c**). Callout charts in **a** show select seasonal profiles in bi-weekly intervals of mean output and primary climate drivers: GHI, RH and temperature. Output (in l d⁻¹ m⁻²) normalized to horizontal device area in sunlight. Real devices will perform below maximum theoretical potentials. Overlaid dot density (red) of 2.2 billion people without SMDW. Placement of dots is spatially arbitrary across the survey region. Produced in ArcGIS 10.

Researchers and device inventors can cross-reference Fig. 4 when making trade-off decisions between sets of technical specifications and servable regions and people. Recent experiments[4,5,37] show rapid improvements in multi-cycled sorption material yield, ranging from 0.1 to more than 8.0 l d⁻¹ kg⁻¹ sorbent in outdoor conditions (RH 10–60%, GHI < 1,000 W m⁻²), and show inflections in performance along similar ranges as population distributions[11,31] (RH 30–50%, GHI 400–600 W m⁻²). Advancements in device efficiencies from innovative design architectures[38] and novel high-performance physical sorbents[15,17,39–41] show promise for increasing SC-AWH output. Individual specific yields from materials experiments or prototypes can be plotted in Fig. 4d for benchmarking against target impacts. Validated device performance in outdoor field conditions and published output tables and are needed for global researchers to advance progress of AWH.

The long-term averaged output of an AWH device is an important but limited metric. Seasonal, weekly and diurnal variability in output will influence user adoption and market viability. Some seasonal profiles are explored in Extended Data Figs. 4–8. Short periods of shortfall may be supplemented by storage from previous surpluses. Rainfall collection or alternative sources would be required for seasonal shortfall periods, such as those in monsoon climates. Use of multiple water sources and seasonal switching are well established in the literature, although there may be trade-offs with respect to water quality and contamination[42,43], reinforcing the need for in-depth knowledge of existing water access practices when deploying AWHs, with a focus on household water treatment and safe storage.

The hydro-ecological impacts of AWH for drinking water are probably negligible given the scale of the global atmospheric water budget. Serving all 2.2 billion people without SMDW at 10 l d⁻¹ sums to approximately 8 km³ yr⁻¹, a mere 0.20% of the net water extraction of global cropland (4,000 km³ yr⁻¹) and 0.01% of total evapo-transpiration over land[44] (65,500 km³ yr⁻¹).

SC-AWH devices have the potential to be low-cost. Most design architectures have few moving parts (for example, a slowly rotating sorbent wheel[8]), and can be constructed from widely available components. Advanced sorbent materials (for example, MOFs or TRP) will need to be mass manufactured to reach cost targets. New high-volume manufacturing methods for MOFs[45,46] have the potential to drastically reduce costs.

Technology development is only one part of the complex problem of safe water access; user-centric formative research with a wide variety of

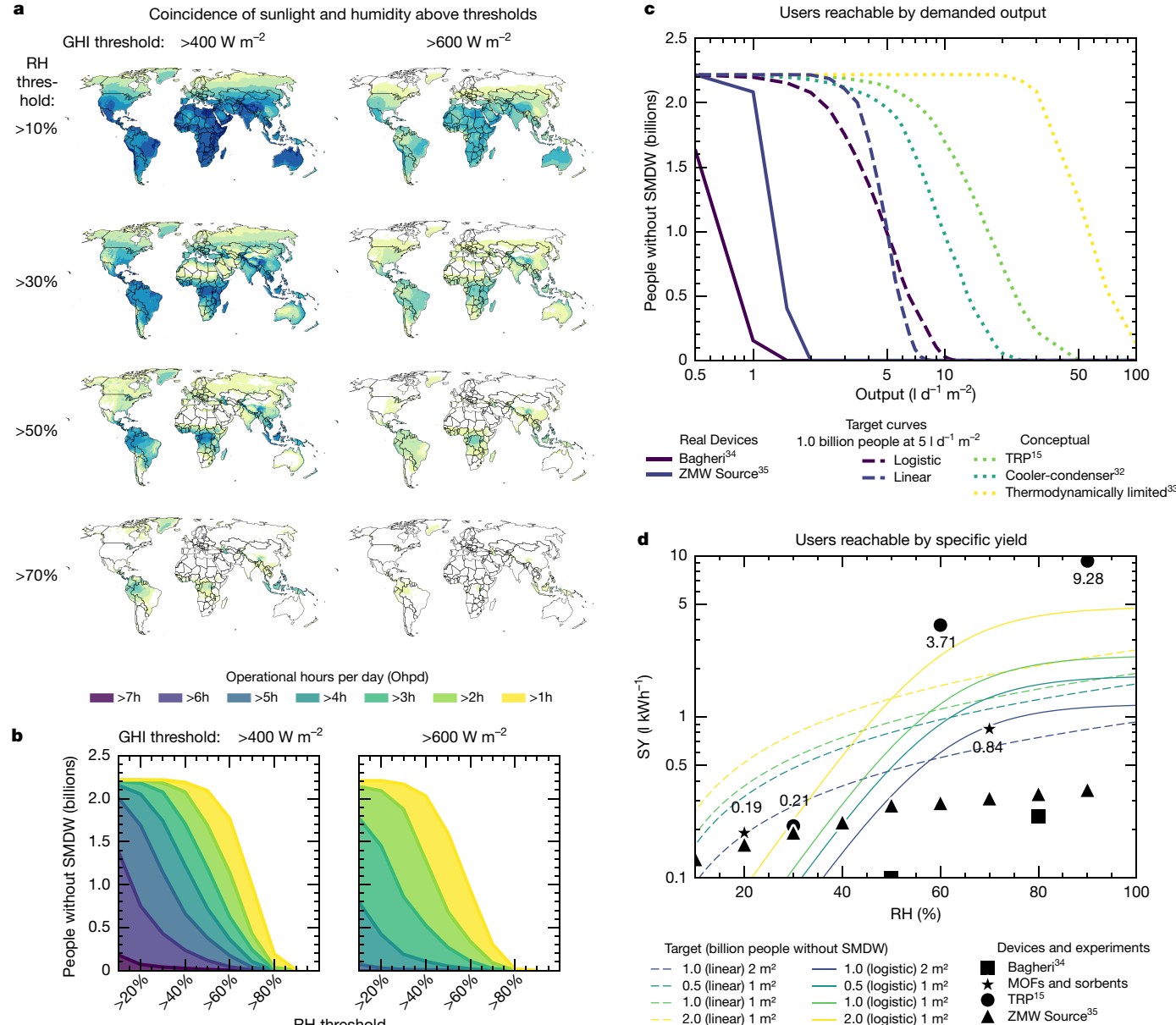

**Fig. 4 | AWH technology parameters in relation to global impact targets.**
**a**, **b**, Geographic distribution (**a**) and sum (**b**) of population without SMDW living in areas meeting parametric thresholds relevant to operation of SC-AWH devices. Operational hours per day (Ophd) is the mean daily duration of both sunlight (GHI) and RH thresholds exceeded simultaneously. Usage example: a device requiring more than 5 h d$^{-1}$ of sunlight above 400 W m$^{-2}$ must operate down to 40% RH to reach approximately 700 million users. **c**, **d**, People without SMDW reachable in relation to mean daily output normalized to horizontal device area in sunlight (**c**) and SY profile (**d**). Target curves are hypothetical SY profiles capable of providing 5 l d$^{-1}$ for a given solar collection area. Water

output and SY targets scale linearly with device area in sunlight. For demonstration we therefore show that, for a given RH, doubling the area of a device from 1 m$^2$ to 2 m$^2$ halves the target SY requirement to achieve SMDW for a target population. ZMW Source profile approximated from the manufacturer's technical specifications sheet[35]. Note that the full ZMW panel is approximately 3 m$^2$. Experimental values for MOFs and sorbents are taken from experiments[3,36] (0.19 l kWh$^{-1}$ and 0.84 l kWh$^{-1}$), and TRP is taken from ref. [15], all converted as in ref. [32]. Values for the Bagheri device[34] assume work instead of heat input; therefore photovoltaic efficiencies were applied when converting from GHI. Maps are produced in ArcGIS 10.

end users is critical for ensuring that devices are adopted widely. Similar to bottled water[21] SC_AWH devices could paradoxically undermine efforts to develop permanent piped infrastructure. Product affordability and adoption require parallel financial and socio-cultural efforts such as scaling availability of loans, promoting awareness of waterborne disease risk and increasing women's influence over community decisions[47–49].

Our analysis demonstrates that daytime climate conditions may in fact be sufficient for continuous-mode AWH operation in world regions with the highest human need. This assessment suggests that focusing device design criteria on maximum impact and reducing

costs of off-grid production of drinking water at the household scale is a worthwhile effort.

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

## Methods

### Water access data processing

Data on drinking water coverage by region was acquired from the WHO/UNICEF JMP. The JMP acts as official custodian of global data on water supply, sanitation and hygiene[2] and assimilates data from administrative data, national census and surveys for individual countries, and maintains a database that can be accessed online through their website. We accessed data tables for national and subnational drinking water service levels from https://washdata.org.

JMP datasets are not geographically linked to official boundary files. We joined the tables to GIS boundaries obtained from the following open-source collections: GADM (https://gadm.org), the Spatial Data Repository of the Demographic and Health Surveys Program of USAID (DHS) and the Global Data Lab of Radboud University (GDL)[2,50–53]. Subnational regions reported by the JMP are unstructured, representing various regional administrative levels (province, state, district and others).

The JMP national and subnational data were joined to GIS boundaries using a custom geoprocessing tool built in Python and ArcGIS 10. The tool joins the available JMP subnational-level survey data to the closest name match of regional boundary names from a merged stack of GADM (admin1, admin2 and admin3), DHS and GDL boundaries worldwide. The JMP national-level survey data is then joined to GADM national (admin0) boundaries for countries which have no subnational data available. Finally, the two boundary-joined datasets (national and subnational) are merged, processed and exported as a seamless global fabric of water-stressed-population data at the highest respective spatial resolutions available (Fig. 1a).

JMP does not report the breakdown between the SMDW and basic service level within subnational regions, and instead reports a combined category called 'at least basic' (ALB). To estimate the SMDW values in subnational regions, a simple cross-multiplication was performed using the splits at the national level:

$$\text{SMDW}_{\text{subnational}} = \frac{\text{SMDW}_{\text{national}}}{\text{ALB}_{\text{national}}} \times \text{ALB}_{\text{subnational}},$$

where $\text{ALB}_{\text{national}}$, $\text{ALB}_{\text{subnational}}$ and $\text{SMDW}_{\text{national}}$ are known values.

Validation of the cross-estimation of share of SMDW from ALB for subnational regions was conducted on a reference dataset of nationally representative household surveys that collected data on all criteria for SMDW[54], shown in Extended Data Fig. 2. We report regression results of $R^2 = 0.87$ and a standard error of 3.67, indicating a bias which over-reports SMDW share and a probable underestimate of people living without SMDW in our study. This discrepancy comes from JMP calculations of SMDW that rely on the minimum value of multiple drinking water service criteria (free from contamination, available when needed and accessible on premise) rather than considering whether individual households meet all criteria for SMDW[55].

The fraction of population without SMDW was multiplied by residential population values in the WorldPop top-down unconstrained global mosaic population count of 2017 at 1 km spatial resolution[56] (https://www.worldpop.org). WorldPop was accessed online as a TIF image and imported to Google Earth Engine. The year 2017 was chosen to more closely match water access data from JMP. The percentages reported by JMP are probably not uniform within most regions[57], introducing an unknown error to Fig. 1b, but represent the best estimate available to us given the limitations of these regionally reported data.

### Climate input and conversion approximations

**GHI and reference plane.** We used GHI (in W m$^{-2}$) as solar energy input data. GHI has good availability in climate datasets and introduces the fewest number of assumptions. Since GHI describes the irradiance in a locally horizontal reference plane, this approximation is only exact for devices having a horizontally oriented solar harvesting area. Annually averaged comparisons between horizontal and optimal fixed-tilt panels show negligible differences in direct plus diffuse radiation in tropical latitudes, and ratios below 25% in locations within 50° north and south latitudes[58]. Those seeking precise absolute predictions for tilted devices or higher latitudes are encouraged to adapt the provided code to their specific assumptions.

**Conversion from SY to AWH output.** As discussed in the main text, solar-driven AWH devices typically have one of two predominant energy inputs: thermal (converted directly from incident sunlight on the device) or electrical (from PV). Here, the energy units used to calculate yield in l kWh$^{-1}$ are incident solar energy directly from GHI. The various assumptions are made in relation to the reported values based on their source. The thermal limits[33], target curves, and experimental results reported by TRP[15] and MOFs were assumed to have direct (100%) conversion from sunlight to heat. For the ZMW device, the table provided by the manufacturer accounts for system losses, so the table values were directly converted in our model[35]. For ref. [34] and the cooler–condenser limits from ref. [32], which both assume work input instead of heat, we applied a typical PV conversion efficiency of 20% to convert from sunlight kWh (GHI) to kWh$_{\text{PV}}$ (electrical work) input to the device[59].

**Sufficiently short sorbent cycling times.** AWH-Geo assumes continuous or quasi-continuous AWH. AWH-Geo considers each 1-h timestep independently and is thus stateless. Aside from edge cases, this is a safe assumption for mass efficient SC-AWH devices, which typically have time constants shorter than 1 h, both for sorbent cycling and for most of the thermal time constants. For devices with longer time constants, batch devices or processes with slow (de)sorption kinetics, this assumption may introduce increased error, and may require further adaptation of the provided code.

### Climate time-series calculation

AWH-Geo is a resource-assessment tool for AWH. It consists of a geospatial processing pipeline for mapping water production (in litres per unit time) around the world of any solar-driven continuous AWH device that can be characterized by an output table of the form output = $f$(RH, $T$, GHI).

Output tables show AWH output values in l h$^{-1}$ or l h$^{-1}$ m$^{-2}$ across permutations of the 3 main climate variables in the following ranges: RH between 0 and 100% in intervals of 10%, GHI between 0 and 1,300 W m$^{-2}$ in intervals of 100 W m$^{-2}$, and $T$ between 0 and 45 °C in intervals of 2.5 °C (2,145 total output values). The tables are converted into a 3D array image in Google Earth Engine and processed across the climate time-series image collection for the period of interest. Finally, these AWH output values are composited (reduced) to a single time-averaged statistic of interest as an image.

Climate time-series data was acquired from the ERA5-Land climate reanalysis from the European Centre for Medium-Range Weather Forecasts (ECMWF)[60], accessed from the Google Earth Engine data catalogue. ERA5-Land surface variables were used in 1-h intervals and 0.1° × 0.1° (nominal 9 km). The 10-year analysis period (2010–2019, inclusive) was used for this work, and represents a period long enough to provide a reasonable correction for medium-term interannual climatic variability.

Climate variables GHI and T were matched to ERA5-Land parameters 'Surface solar radiation downwards' (converted from cumulative to mean hourly) and '2 metre temperature' (converted from K to °C), respectively. RH was calculated from the ambient and dew point temperature parameters in a relationship derived from the August–Roche–Magnus approximation[61] rearranged as:

$$\text{RH} = 100\% \times \frac{\text{e}^{\left(\frac{aT_{\text{d}}}{b+T_{\text{d}}}\right)}}{\text{e}^{\left(\frac{aT}{b+T}\right)}}$$

where $a$ is 17.625 (constant), $b$ is 243.04 (constant), $T$ is the ERA5-Land parameter '2 metre temperature' converted from K to °C, and $T_d$ is the ERA5-Land parameter '2 metre dewpoint temperature' converted from K to °C.

Spot validation of the climate parameters and the mapped output was performed manually in Google Earth Engine across several timesteps in 2016 in Ames, Iowa (using the Iowa Environmental Mesonet AMES-8-WSW station[62]) and showed insignificant error (< 5%).

## Mapping upper bounds

Figure 3a maps thermodynamic upper bound outputs for SC-AWH based on an equation from Kim et al. [33], reproduced below.

$$\frac{\dot{Q}_{hot,in,min}}{\dot{m}_{water,out}} = \left[ \frac{1}{\omega_{air,in} - \omega_{air,out}} (e_{air,out} - e_{air,in}) + e_{water,out} \right] \times \left( 1 - \frac{T_{ambient}}{T_{hot}} \right)^{-1}$$

where $\dot{Q}_{hot,in,min}$ is the minimum input heat flux (in $W_{heat}$) required to drive the process, $T_{hot}$ is the temperature (in K) at which the input heat is delivered, $T_{ambient}$ is the ambient temperature (in K) at which heat is rejected and water and air exit the process, $\dot{m}_{water,out}$ is the production rate of liquid water by mass, $\omega$ denotes humidity ratios in kg of water per kg of dry air, $e$ denotes specific exergies, which can be looked up for given temperatures and humidities, subscript air,in denotes ambient air drawn in at $T_{ambient}$ from which to extract moisture, subscript air,out denotes air exiting the process at $T_{ambient}$ after extracting some moisture from it, subscript water,out denotes liquid water exiting the process at $T_{ambient}$ as the desired product.

Parameters not present in this formula, but that are in Kim's underlying derivation: this upper limit is obtained for a small recovery ratio (RR ~ 0) chosen for numerical stability and for reversible process conditions (entropy generation, $S_{gen} = 0$).

Kim's model assumes an AWH in which the fundamental energies required are driven by input heat supplied at a temperature $T_{hot}$. The limit it represents applies independent of the process, number of stages, sorbent choice, and so on, as long as heat drives the process.

We adapt Kim's model to solar energy input, assuming an idealized conversion efficiency from solar irradiance to usable heat of 100%. This idealization retains a robust upper bound without bringing in additional parameters. Literature values for theoretical sun-to-heat efficiency limits range from >99.99 to 95.80% for thermal absorbers, depending on the level of angular selectivity[63].

Rearranged, Kim's model yields

$$\frac{\dot{V}_{water,out}}{A} \leq E_{GHI} \times \left( 1 - \frac{T_{ambient}}{T_{hot}} \right) \times \left[ \frac{1}{\omega_{air,in} - \omega_{air,out}} (e_{air,out} - e_{air,in}) + e_{water,out} \right]^{-1} \times \frac{1}{\rho_{water}}$$

where, in addition, $\dot{V}_{water,out}$ is the production rate of liquid water by volume, $A$ is the area harvesting sunlight (see approximation section below), $E_{GHI}$ is GHI in $W_{sun}\,m^{-2}$, and $\rho_{water}$ is the density of water.

This is now a function of the three key climate variables: GHI (in the first term), ambient temperature (in the second and hidden in the third term) and RH (entering the third term). This was converted to an output table and processed through the AWH-Geo pipeline and presented in Fig. 3a. While this can be run for any choice of parameter $T_{hot}$, we present figures here for $T_{hot} = 100$ °C, a temperature still achievable in low-cost (non-vacuum) practical devices without tracking or sunlight concentration. Higher driving temperatures increase the upper bound for water output. For the limits analysis, values of RH above 90% are clamped to prevent unrealistically high theoretical outputs as Kim's

equation goes to infinity at 100% RH. A further assumption is made that new ambient air is efficiently refreshed.

Figure 3b maps the maximum yield for active cooler–condensers without recuperation of sensible heat—all given work input and an optimum coefficient of performance of the cooling unit at a condenser temperature that maximizes specific yield as modelled by Peeters[32], which we digitized from their fig. 11. Peeters chose to set yield to zero whenever frost formation would be expected on the condenser. Since Peeters assumes work input, we convert from solar energy (GHI) to $kWh_{PV}$ as discussed above.

Figure 3c maps Zhao's experimental results from a TRP using a logistic regression curve fit to their reported SYs of 0.21, 3.71 and 9.28 l kWh$^{-1}$ at 30, 60 and 90% RH, respectively[15]. The terms of the curve fit are reported in the next section.

Custom yellow to blue map colours are based on www.ColorBrewer.org, by C. A. Brewer, Penn State[64].

## Specific yield and target curves

Two simple characteristic equations, linear and logistic, were used to fit a limited set of SY and RH pairs from laboratory experiments or reported values and plotted through AWH-Geo using calculated output tables. Hypothetical curves of similar form whose terms were adjusted iteratively in AWH-Geo to goal-seek a target output (5 l d$^{-1}$) and user base, and are reported here (for 1-m$^2$ devices). In the following equations, RH in % is taken as a fraction (for example 55% is equivalent to 0.55).

The linear target curve is a simple linear function which crosses the $y$-axis at zero:

$$SY(RH) = a \times RH$$

where $a$ is set to 1.60, 1.86 and 2.60 L/kWh to reach targets of 0.5, 1.0, and 2.0 billion people without SMDW, respectively, and RH is input RH (fractional).

The logistic target curve is a logistic function:

$$SY(RH) = \frac{L}{1 + e^{-k(RH-RH_0)}}$$

where $L$ is set to 1.80, 2.40 and 4.80 L kWh$^{-1}$ to reach targets of 0.5, 1.0 and 2.0 billion people without SMDW, respectively, $k$ is the growth rate set to 10.0, and RH and $RH_0$ are input RH (fractional), and 0.60, respectively.

The SY values reported by Zhao for TRPs (which they term 'SMAG') were fit to a logistic function of the same form with the following parameters: $L$ set to 9.81 L kWh$^{-1}$, $k$ set to 11.25 and $RH_0$ set to 0.645.

The resulting fitted SY profile is expanded into an output table. As with all reports providing SY values instead of full output tables, this forces an assumption of linearity in heat rate (approximately equal to GHI), which may introduce error at lower GHI levels. Zhao reports SY of the TRP material is consistent across temperature below 40 °C—the material's lower critical solution temperature—above which its performance drops precipitously. Accordingly, we set the SY to 0 l kWh$^{-1}$ for temperatures ≥40 °C in the output table.

Bagheri reported performance of three existing AWH devices across several climate conditions using an 'energy consumption rate' in kWh/L, which can be considered to be the SEC, and the simple reciprocal of SY. Instead of fitting a logistic curve to the reciprocals, we fit an exponential function to the average SEC of the three devices in conditions above 20 °C of the equation:

$$SEC(RH) = 9.03 e^{-2.99RH}$$

where SEC is specific energy consumption in $kWh_{PV}\,l^{-1}$ and RH is fractional.

This was applied to RH and taken as reciprocal in an output table and run through AWH-Geo. Since Bagheri reports the equivalent of $kWh_{PV}$, we scale to adapt to GHI input with a photovoltaic conversion efficiency as discussed above.

For performance of the ZMW device (the company's ~3 $m^2$ SOURCE Hydropanel), we used values from the panel production contour plot in the technical specification sheet available from the manufacturer's website[35]. The decision for inclusion was made owing to the importance as an early example of a SC-AWH product with commercial intent. Values in l per panel per day were taken at each 10% RH step at 5 kWh $m^{-2}$, assumed to represent kWh $m^{-2}$ $d^{-1}$, and divided by 15 kWh (~3 $m^2 \times 5$ kWh $m^{-2}$) to convert to SY in l kWh$^{-1}$. From the resulting SY curve, an output table was generated and processed with AWH-Geo.

## Coincidence analysis and population sums

The coincidence analysis was run through AWH-Geo across 70 threshold pairs given the full permutation set of RH from 10 to 100% and GHI from 400 to 700 W $m^{-2}$ threshold intervals, using binary image time series. The resulting mean multiplied by 24 represents average hours per day thresholds are met simultaneously, giving ophd. Below is a functional representation of this time-series calculation:

$$\langle (RH_{t,px} > RH_{threshold}) \&\&_{simultaneous} (GHI_{t,px} > GHI_{threshold}) \rangle_{time\ average}$$

where $RH_{t,px}$ is the RH in the map pixel px at time $t$, $RH_{threshold}$ is the threshold of RH above which the device is assumed to operate, $GHI_{t,px}$ is the GHI in the map pixel px at time $t$, and $GHI_{threshold}$ is the threshold of GHI above which the device is assumed to operate.

The population calculation was then conducted on these images in Google Earth Engine.

Zonal statistics were performed on the mean ophd images as integers (0–24) using a grouped image reduction (at 1,000-m scale) summing the population integer counts on the population without SMDW distribution image created previously (derived from WorldPop). This reduction was performed at 1,000 m. Validation was performed in Google Earth Engine on single countries within single ophd zones and showed insignificant error (<2%). The population results were collected as a table (feature collection) and population was summed cumulatively within stacked ophd zones. These were exported to R for plotting in Fig. 4b.

To assess the sensitivity of results to the choice of climate and population dataset, we performed a coincidence analysis (Fig. 4b) with alternative datasets and provide those results in Extended Data Fig. 1.

As an alternative climate dataset to ERA-5 (1 h, 9 km), we used NASA's Global Land Data Assimilation System (GLDAS) 2.1 at 0.25° × 0.25° spatial resolution (nominally 30 km) and 3 h temporal resolution[65] during the period concurrent with the main results, 2010–2019. As an alternative population dataset to WorldPop 2017, we used Oak Ridge National Laboratory's LandScan 2017 ambient population counts at 1 km spatial resolution[66]. Two results comparisons were calculated: (1) GLDAS calculated with WorldPop 2017 for direct comparison of climate data input, and (2) GLDAS calculated with LandScan for comparison of climate and population dataset substitution.

The intercomparisons suggest there is negligible sensitivity to the population dataset used, but substantial and systematic sensitivity to the climate dataset used, while all intercomparisons agree in main features and qualitative conclusions. The spatially and temporally (3×) coarser GLDAS dataset consistently results in lower predictions of water output and impact than the finer ERA-5 climate reanalysis. We speculate that the 3-h timesteps of GLDAS are insufficient to capture the performance-critical humidity and GHI dynamics throughout the day (probably morning and evening hours), and, similarly, the 30-km pixels are insufficient to resolve fine-scale climate patterns driven by topographic and other microscale physiographic effects. This illustrates the importance of using high-resolution climate datasets.

## Variability statistics of AWH output

To go beyond annual averages and study availability, we introduce a set of metrics we named moving average density 90th percentile (MADP90).

The MADP90-t represents a device's average output rate (l $d^{-1}$ $m^{-2}$) that will be exceeded for 90% of periods lasting $t$ days at the given location. MADP90 is calculated from the derived P90 value across a probability density function (PDF) of daily mean output during each $t$-day window in the time series (2010–2019). The result is a scalar that can be mapped spatially. Moving-window periods of 1, 7, 30, 60, 90 and 180 days were examined in this study. MADP90-results are available as additional results and map layers in AWH-Geo.

Extended Data Fig. 3 provides an example set of PDFs for a location in southwest Tanzania. Each of the P90 values correspond to a version of the MADP90 metric corresponding to a moving window period. The P90 value naturally increases with $t$ in most geographic locations as the PDF tightens its dispersion about the natural (P50) mean.

## Data availability

The software and datasets generated during and/or analysed during the current study are available in the following repositories. GitHub: https://github.com/AWH-GlobalPotential-X/AWH-Geo; Figshare: https://doi.org/10.6084/m9.figshare.c.5642992.v1; JMP Geoprocessor package (Python and ArcGIS geoprocessing model); JMP Geofabric dataset (shapefile); population without SMDW image data layer (geoTiff); upper limit AWH output data layers (geoTiff); coincidence analysis results data tables (Sheets); and output tables used in this study (Sheets). Source data are provided with this paper.

## Code availability

The software used during the current study is available as follows. GitHub: https://github.com/AWH-GlobalPotential-X/AWH-Geo; AWH-Geo application: processor and output viewer with source code; population and result data processing scripts.

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

**Acknowledgements** We acknowledge contributions from many colleagues, including A. Aron-Gilat, D. Youmans, G.L. Whiting, M. Eisaman, S. Lin, J. Sargent, S. McAlister, S. Chariyasatit, B. Dixon, E. St Jean Duggan, F. Carlsvi, K. Stratton, M. McCoy, R. Hessmer, J. Hanna, H. Riley, P. Watson, M. Day, B. Quintanilla-Whye, A. Ramadan, A. Little and D. Moufarege. We thank the WHO/UNICEF JMP team for guidance on drinking water service estimates, in particular T. Slaymaker, R. Johnston and F. Mitis; the team at Google Earth Engine, in particular S. Ilyushchenko, S. Agarwal, T. Erickson, N. Gorelick, M. Hancher, M. Dixon, M. DeWitt, J. Conkling, N. Clinton, K. Reid, E. Engle, W. Rucklidge and the entire Earth Engine development community for advice; C. Caywood for code review; B. Schillings and J. Gagne for internal sponsorship at X. Funding was provided by Google LLC.

**Author contributions** P.H.S. and J.L. conceived the study. J.L., P.D., T.M. and N.T. performed analysis and plots. A.T., N.T., J.L., P.H.S., R.B. and C.H.B. developed arguments. J.L., P.H.S., A.T. and R.B. wrote the paper. This study was conducted as a subset of a larger effort at X, led by P.H.S., M.F., N.T. and A.T., to develop a household AWH as a commercial product, which informed the current study: M.F., N.T. and S.W. led prototype development and experimentation, C.H.B. conducted physical modelling, M.F., S.W., C.T., C.L. and others built devices and conducted experiments, A.T., J.F. and N.K. conducted market and user research.

**Competing interests** We disclose the following potential competing interests. This work was funded by X, The Moonshot Factory (formerly known as Google[x]). X is a part of Alphabet. Both are for-profit entities. X has filed for patent protection for water-from-air devices, on which multiple authors are listed as inventors. Water-from-air devices may represent significant commercial opportunities upon meeting certain metrics. This work may be pursued further in various ways, including as a possible spinout company in which one or more authors may become founders, officers, shareholders, employees or otherwise involved with a financial interest.

**Additional information**
**Correspondence and requests for materials** should be addressed to Jackson Lord or Philipp H. Schmaelzle.

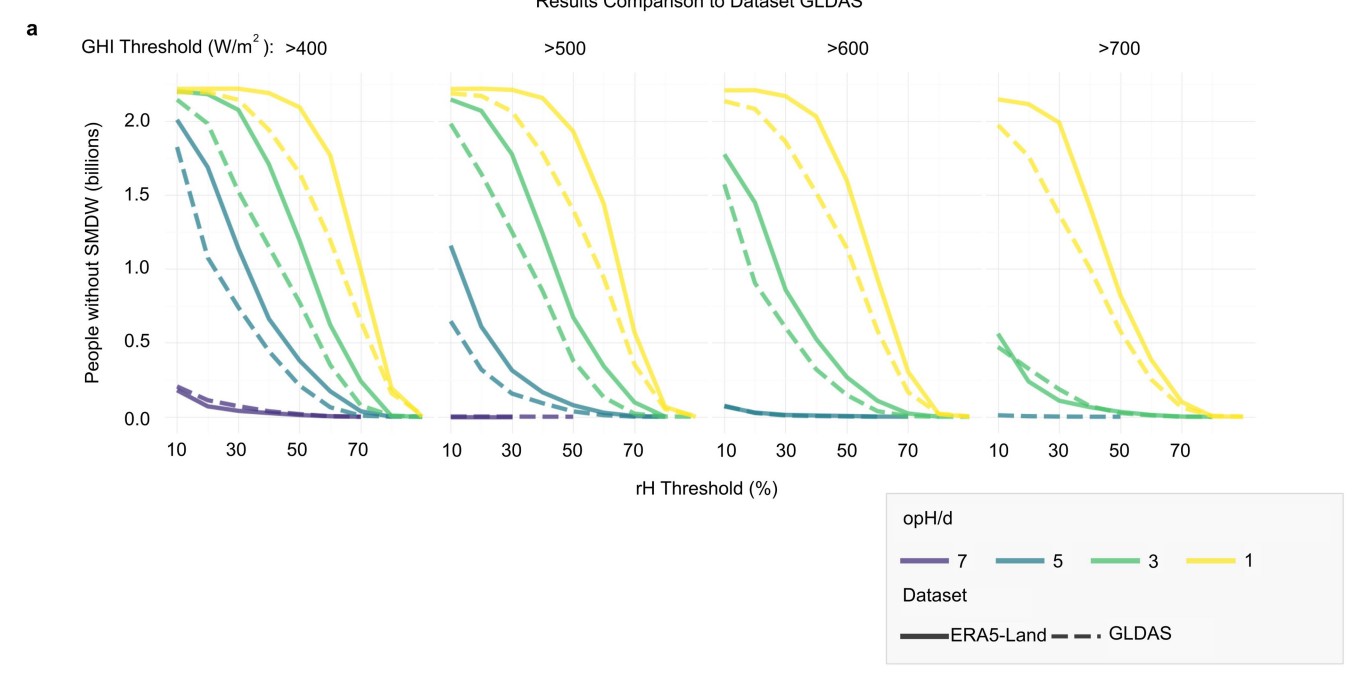

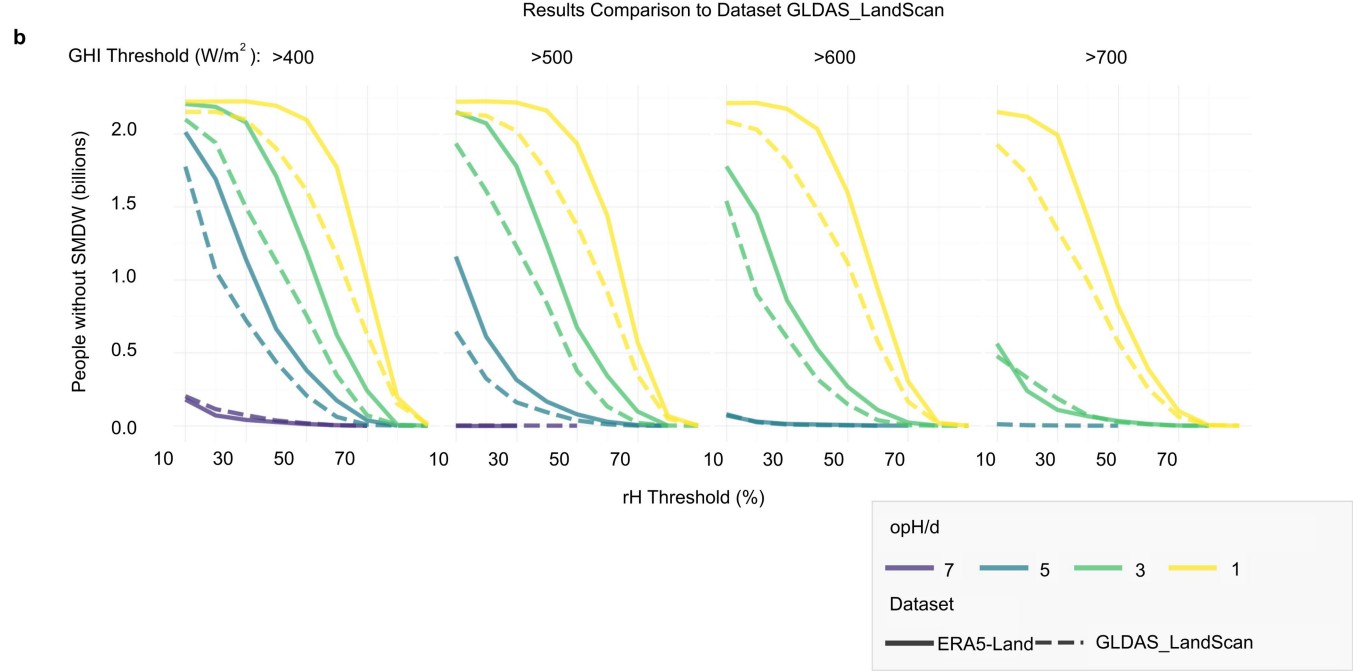

**Extended Data Fig. 1 | Comparison of coincidence analysis results to input datasets.** Main results from coincidence analysis (Fig. 4b, people without SMDW served by *opH/d* of coincident climate threshold) with ERA5-Land and WorldPop 2017 datasets compared with results from **(a)** GLDAS 2.1 climate and WorldPop 2017 population, and **(b)** GLDAS 2.1 climate and LandScan 2017 population datasets. Operational hours per day (*opH/d*) shown across global horizontal irradiance (*GHI*) and relative humidity (*rH*) thresholds.

**a**

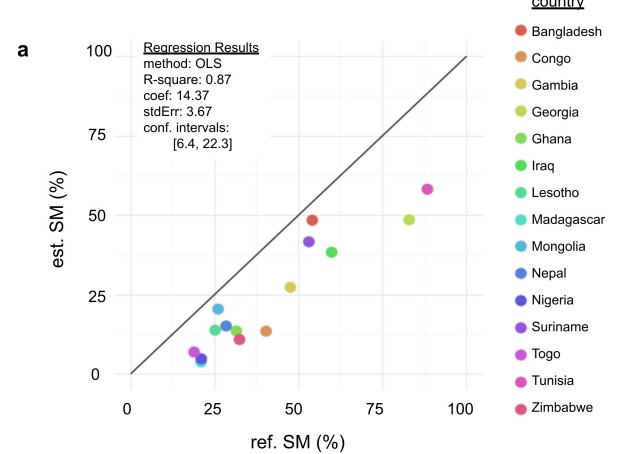

Regression Results
method: OLS
R-square: 0.87
coef: 14.37
stdErr: 3.67
conf. intervals:
[6.4, 22.3]

country
- Bangladesh
- Congo
- Gambia
- Georgia
- Ghana
- Iraq
- Lesotho
- Madagascar
- Mongolia
- Nepal
- Nigeria
- Suriname
- Togo
- Tunisia
- Zimbabwe

**b**

| GHI (W/m²) | opH/d | rH (%) | | | | |
|---|---|---|---|---|---|---|
| | | 10 | 30 | 50 | 70 | 90 |
| | | People without SMDW (millions) | | | | |
| 400 | 1 | 3,045 | 3,047 | 2,887 | 1,348 | 1 |
| 400 | 3 | 3,027 | 2,868 | 1,646 | 326 | 0 |
| 400 | 5 | 2,774 | 1,609 | 516 | 50 | 0 |
| 400 | 7 | 214 | 48 | 14 | 0 | 0 |
| 600 | 1 | 3,035 | 2,984 | 2,176 | 418 | 0 |
| 600 | 3 | 2,482 | 1,198 | 361 | 31 | 0 |
| 600 | 5 | 85 | 15 | 6 | 0 | 0 |
| 600 | 7 | 0 | 0 | 0 | 0 | 0 |

**Extended Data Fig. 2 | Validation of SMDW using household surveys reporting SMDW at household-level. (a)** Charted and **(b)** tabulated validation of cross-estimation of percentage safely managed (SM) from at least basic (ALB) drinking water ladders at sub-national (SN) level from national (N) breakdowns using known reference data set at SN level from WHO/UNICEF JMP data. Reference values from nationally representative Multiple Indicator Cluster Surveys integrating water quality testing (ref. SM) compared with our estimates from the JMP Geoprocessor combining JMP sub-national estimates for ALB and national estimates for safely managed drinking water services (est. SM). Ordinary least squares regression (OLS) resulted in standard error (stdErr) as reported. Sample size *n* = 15. Table **(b)** shows main results (ERA5-Land) population counts after adjustment from regression. Population without safely managed drinking water (SMDW) shown across global horizontal irradiance (*GHI*) and relative humidity (*rH*) thresholds.

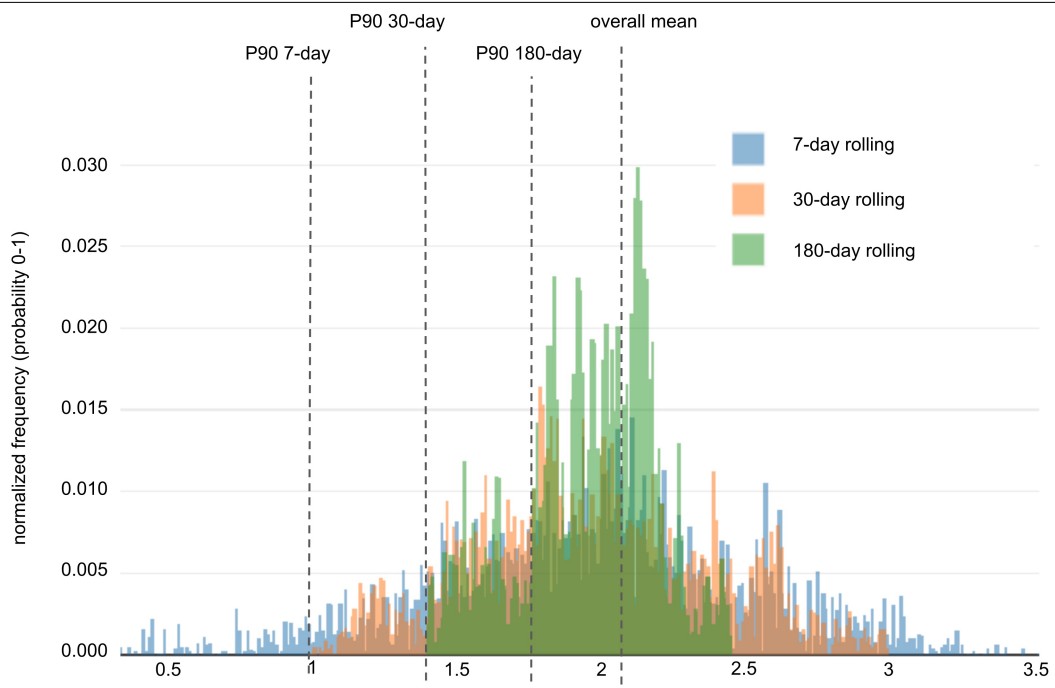

**Extended Data Fig. 3 | Visual representation of MADP90 concept from location in Tanzania.** Histograms of moving-averaged output (L/d/m²) across window periods (indicated in days) for a location in Manda, Tanzania. P90 availability value increases as averaging window period increases. P90 values are estimated and for illustrative purposes only.

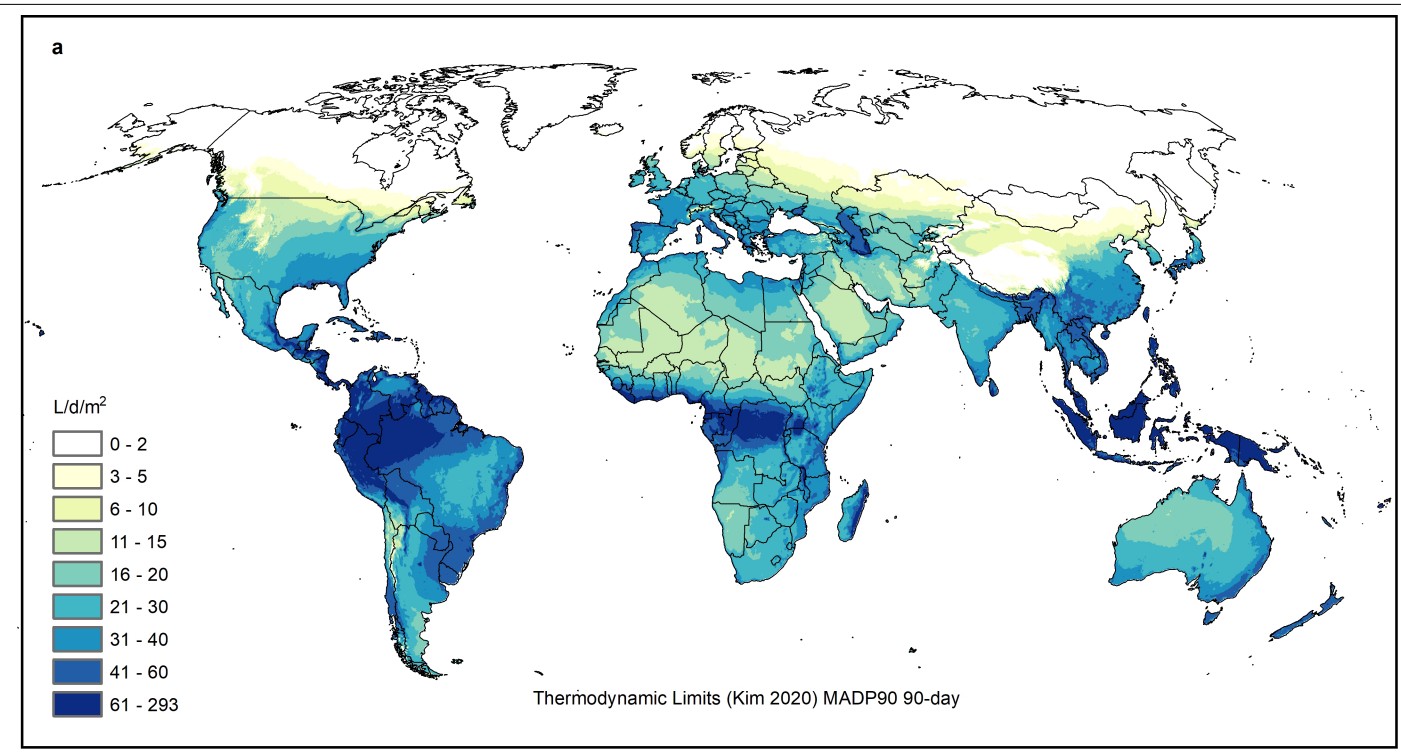

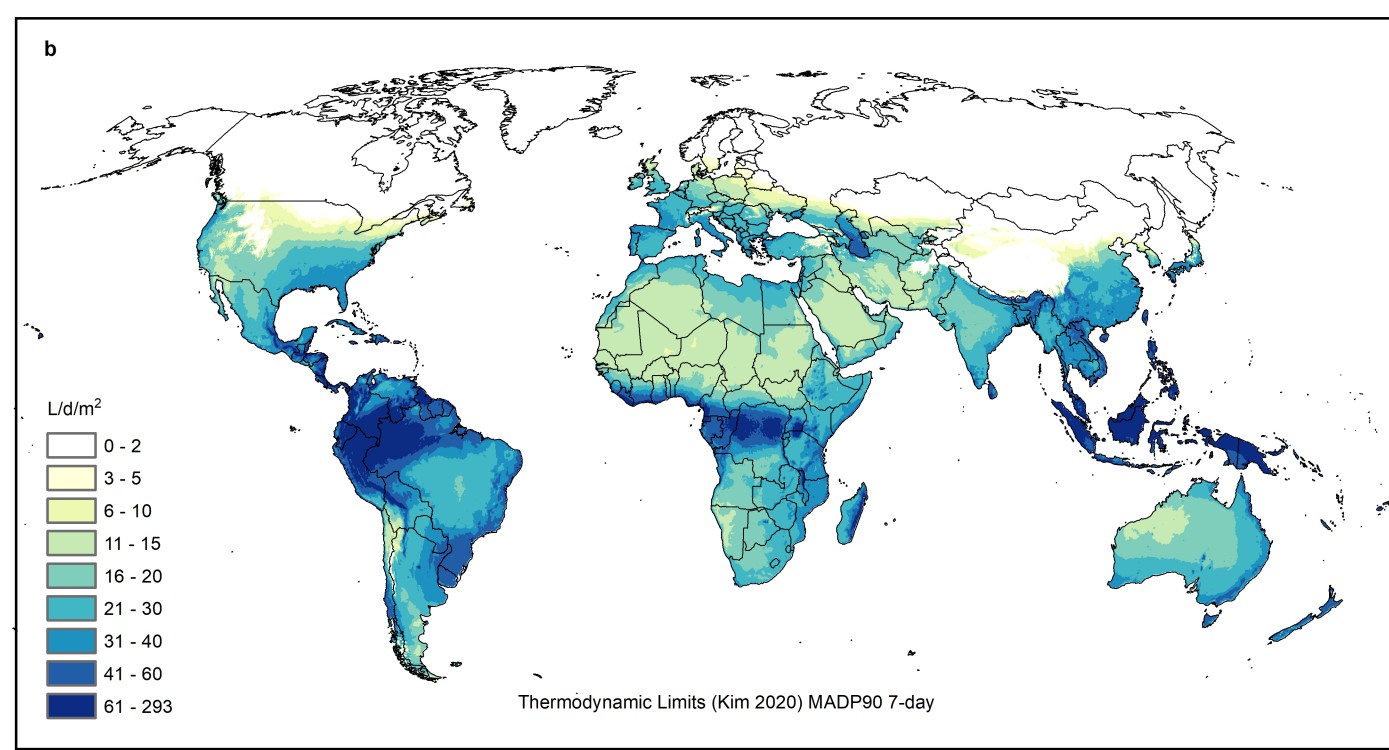

**Extended Data Fig. 4 | Select MADP90 metrics of AWH upper bounds. (a)** MADP90-90day, and **(b)** MADP90-7day values (measure of availability through time) globally for AWH thermodynamic upper bounds (Kim 2020), during ten year 2010–2019 (inclusive) analysis period.

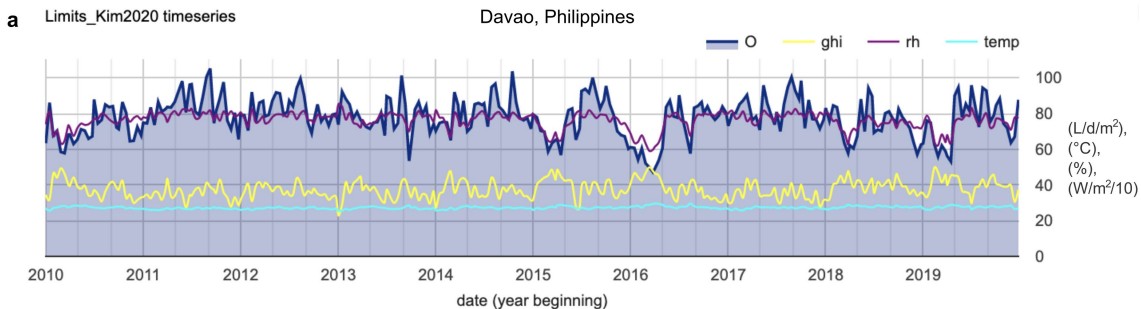

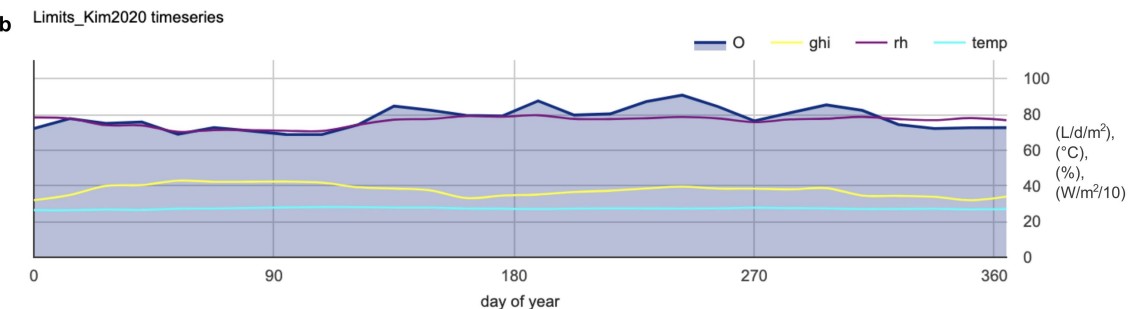

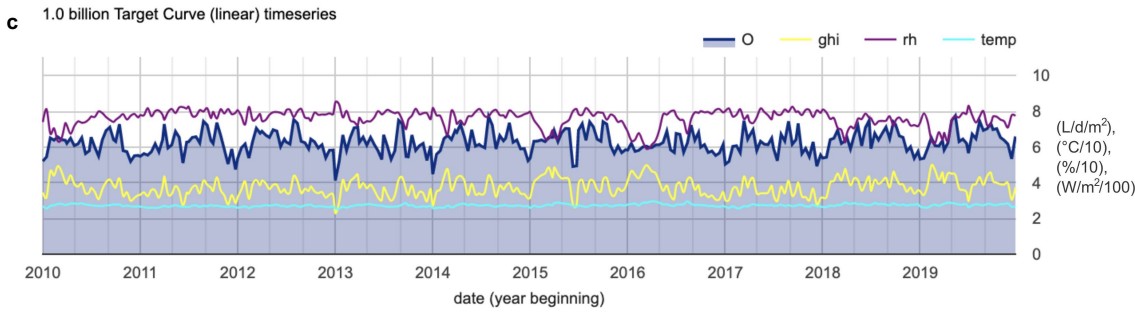

**Extended Data Fig. 5 | Bi-weekly timeseries of AWH output and climate drivers for equatorial profile in Davao, Philippines.** Bi-weekly mean output (L/d/m²), and climate inputs global horizontal irradiance (*GHI*, plotted from 0–1000 W/m²), relative humidity (*rH*, plotted from 0–100 %), and temperature (plotted from 0–100 °C) of **(a)** AWH thermodynamic upper bounds (Kim 2020) during ten year 2010–2019 (inclusive) analysis period for each bi-weekly interval and **(b)** averaged by bi-weekly period annually during this period, and **(c)** for the 1 billion user linear target curve for each bi-weekly interval. Example of a steady, low variability output profile characteristic of equatorial tropics.

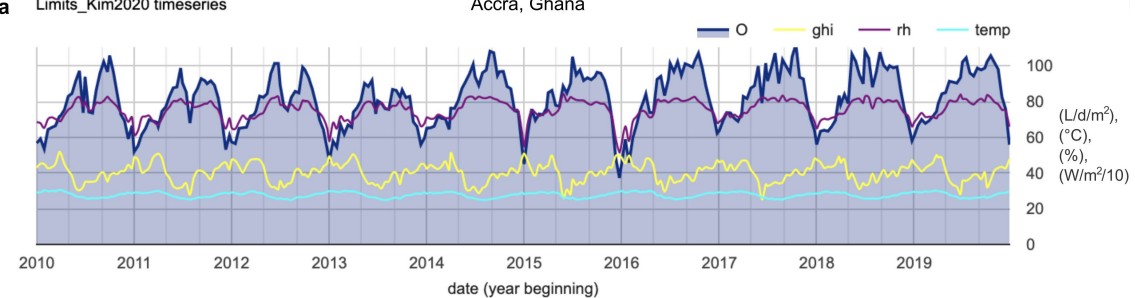

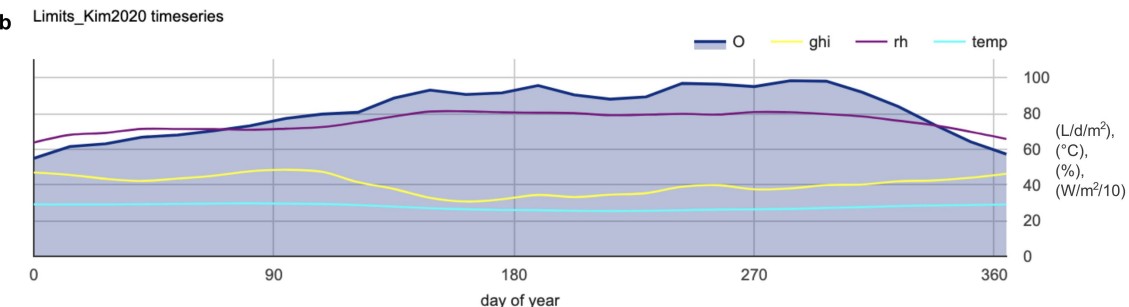

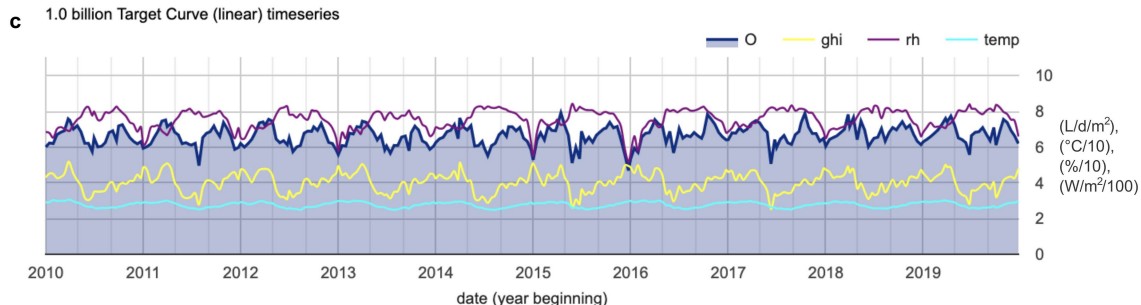

**Extended Data Fig. 6 | Bi-weekly timeseries of AWH output and climate drivers for tropical savanna profile in Accra, Ghana.** Bi-weekly mean output (L/d/m²), and climate inputs global horizontal irradiance (*GHI*, plotted from 0–1000 W/m²), relative humidity (*rH*, plotted from 0–100 %), and temperature (plotted from 0–100 °C) of **(a)** AWH thermodynamic upper bounds (Kim 2020) during ten year 2010–2019 (inclusive) analysis period for each bi-weekly interval and **(b)** averaged by bi-weekly period annually during this period, and **(c)** for the 1 billion user linear target curve for each bi-weekly interval. Example of a seasonal wet-dry tropical savanna climate with moderate semi-annual fluctuations of AWH output driven by *rH*.

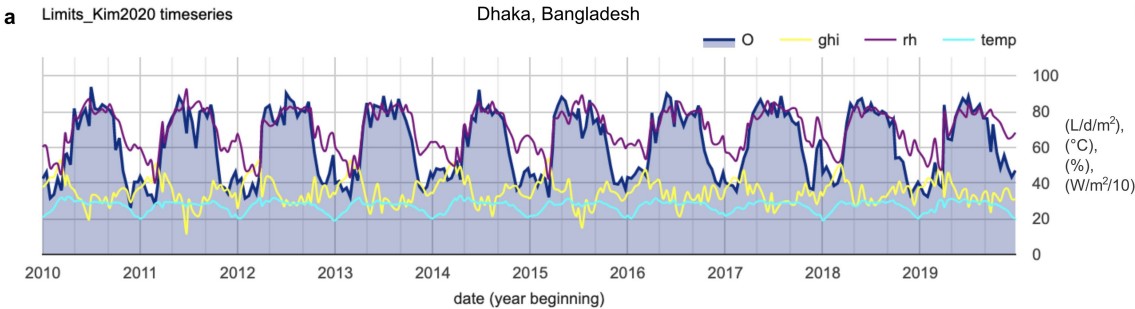

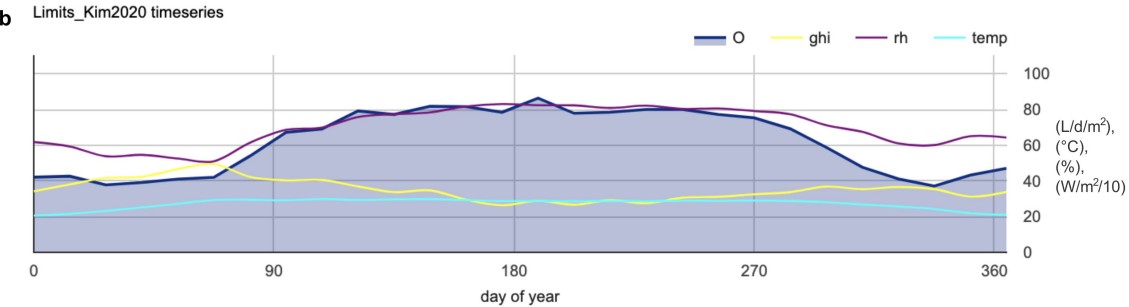

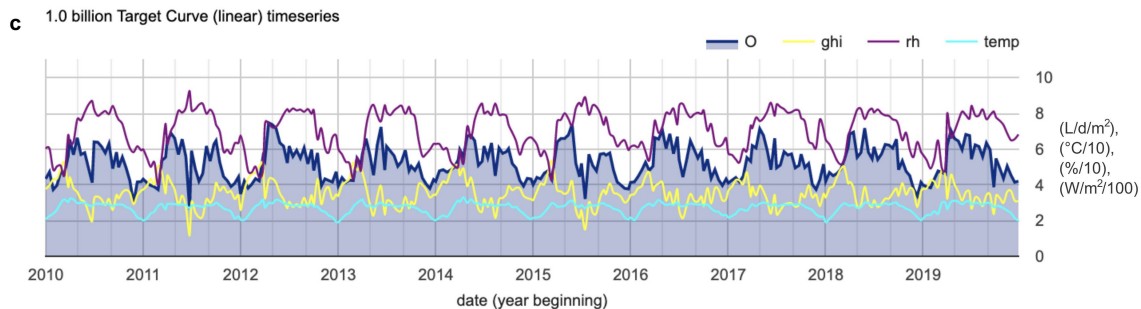

**Extended Data Fig. 7 | Bi-weekly timeseries of AWH output and climate drivers for tropical savanna profile in Dhaka, Bangladesh.** Bi-weekly mean output (L/d/m²), and climate inputs global horizontal irradiance (*GHI*, plotted from 0–1000 W/m²), relative humidity (*rH*, plotted from 0–100 %), and temperature (plotted from 0–100 °C) of **(a)** AWH thermodynamic upper bounds (Kim 2020) during ten year 2010–2019 (inclusive) analysis period for each bi-weekly interval and **(b)** averaged by bi-weekly period annually during this period, and **(c)** for the 1 billion user linear target curve for each bi-weekly interval. Example of a seasonal wet-dry tropical savanna climate with pronounced semi-annual fluctuations of AWH output driven by *rH*.

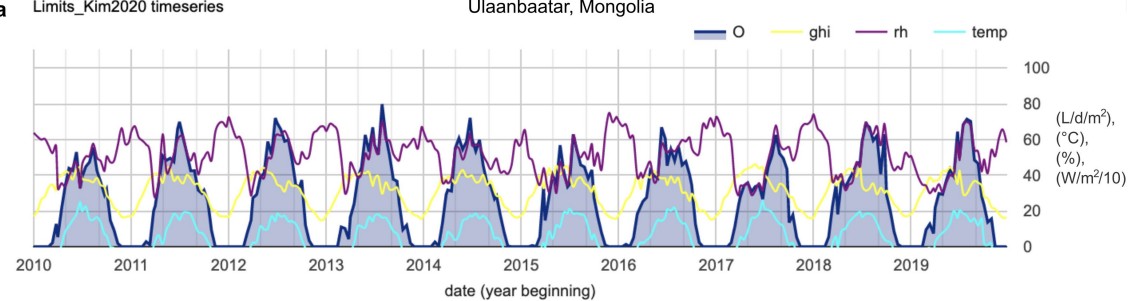

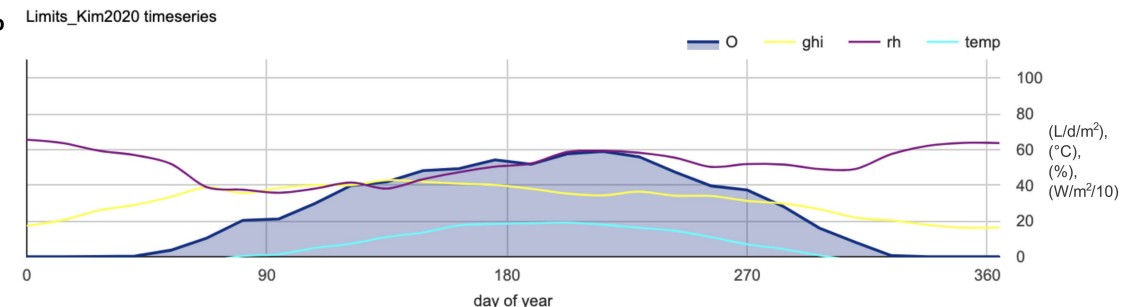

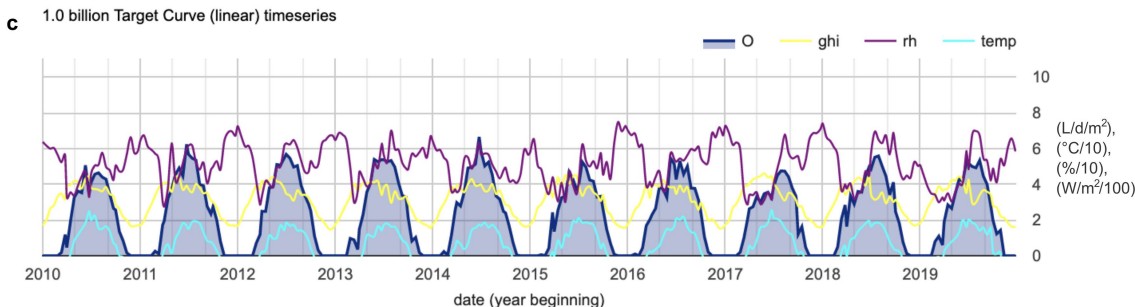

**Extended Data Fig. 8 | Bi-weekly timeseries of AWH output and climate drivers for mid-latitude profile in Ulaanbaatar, Mongolia.** Bi-weekly mean output (L/d/m²), and climate inputs global horizontal irradiance (*GHI*, plotted from 0–1000 W/m²), relative humidity (*rH*, plotted from 0–100 %), and temperature (plotted from 0–100 °C) of **(a)** AWH thermodynamic upper bounds (Kim 2020) during ten year 2010–2019 (inclusive) analysis period for each bi-weekly interval and **(b)** averaged by bi-weekly period annually during this period, and **(c)** for the 1 billion user linear target curve for each bi-weekly interval. Example of a mid-latitude climate with pronounced semi-annual fluctuations of AWH output driven by temperature.

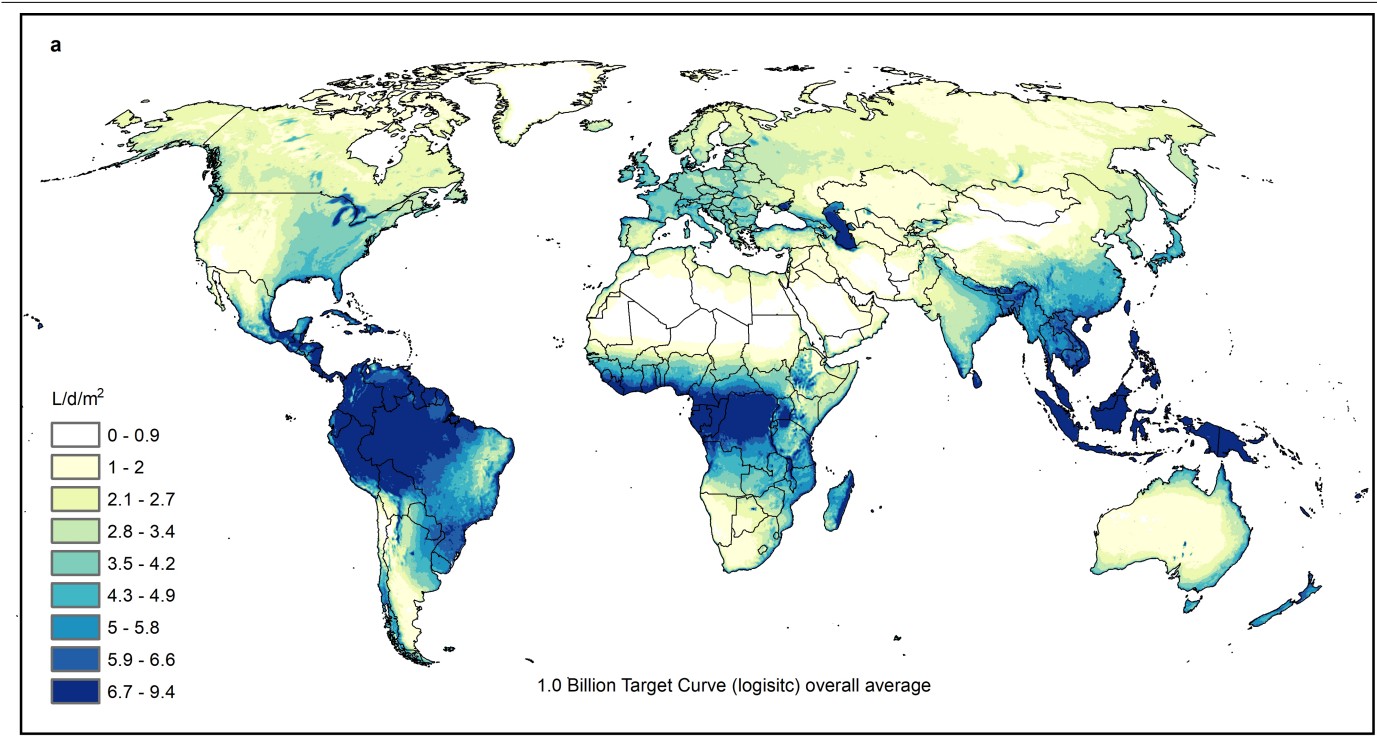

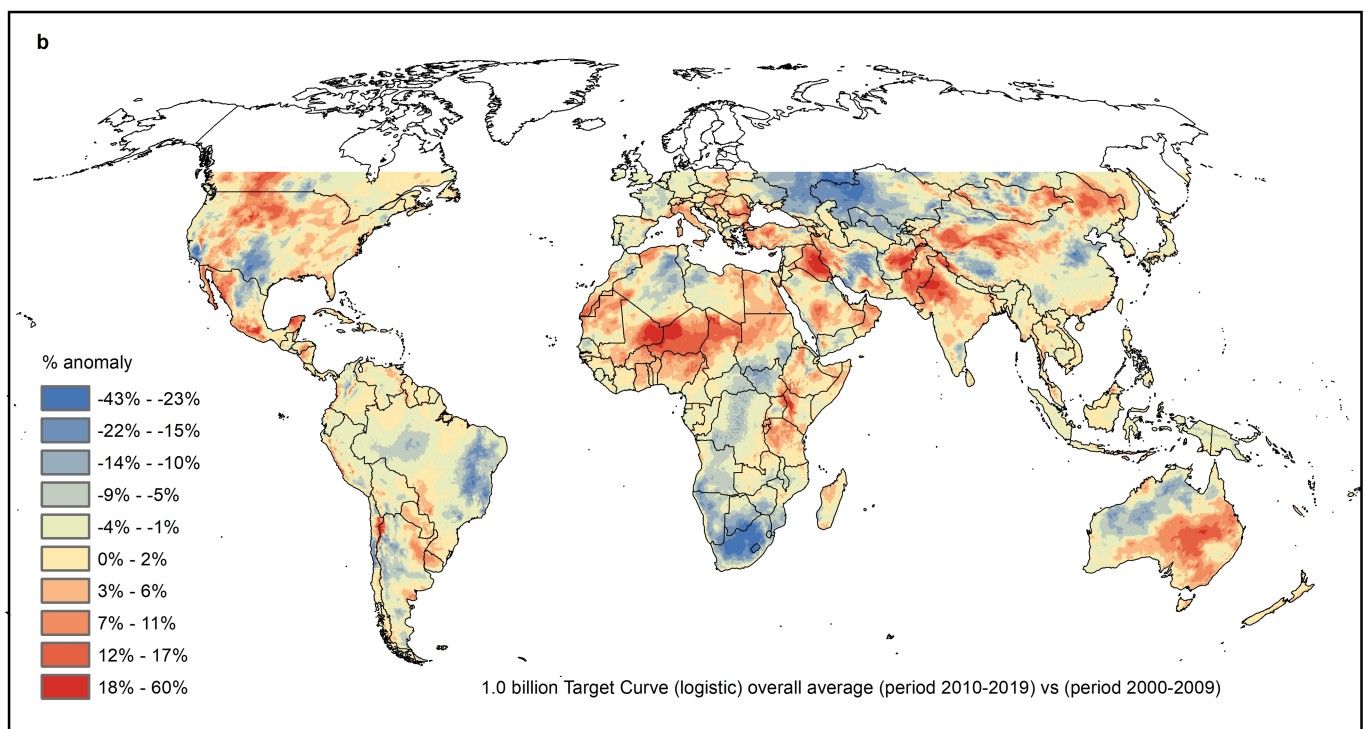

**Extended Data Fig. 9 | Decadal anomaly of AWH output with logistic *SY* profile between 2000–2009 and 2010–2019. (a)** Overall mean output (L/d/m²) of 1 billion user target logistic curve at 5 L/d/m² during ten year 2010–2019 (inclusive) period. **(b)** Ratio (%) anomaly of output of same specific yield (*SY*, in L/kWh) profile averaged over ten year 2000–2009 (inclusive) period. Red colors indicate increasing AWH output with time between the two decades. Blue colors indicate decreasing AWH output.