## [Peer Review File · Nature]

Manuscript Title: Global Potential for Harvesting Drinking Water from Air using Solar Energy

Editorial Notes:

Reviewer Comments & Author Rebuttals

Reviewer Reports on the Initial Version:

Referee #1 (Remarks to the Author):

Review of "Global Potential for Harvesting Drinking Water from Air using Solar Energy" (manuscript number 2020-12-21648B) by Jackson Lord et al.

This manuscript describes a new model which uses specific energy consumption performance data from atmospheric water harvesting (AWH) devices combined with global climate data and solar energy availability (global horizontal irradiance, GHI) to determine potential global water harvesting rates using solar energy. The model also uses data about population without safely managed drinking water (SMDW) to determine the global population whose drinking water source could be met using AWH.

This work is interesting because it is the most comprehensive that I am aware of which provides a continuous global assessment of the potential of AWH based on global climate data and device performance. The work uses specific energy consumption from real commercial devices, devices in literature, and theoretical thermodynamic minimum specific energy consumption. It creates a framework which can be used to assess suitability and potential of AWH for different regions and comparison of devices. However, there are some inconsistencies which must be clarified. First, the treatment of the solar energy input, GHI, must be addressed more clearly. When comparing the performance of the various devices in the paper it is unclear where GHI is being converted from primary energy to electricity (via photovoltaic) and where it is being used in solar-thermal devices. There is also a lack of clarity and consistency throughout the paper and in the presentation of the supplemental information, as I detail further in my comments outlined below. I recommend this manuscript for publication in Nature after the authors address the following comments.

1. In the paper it is unclear how the GHI is treated to convert device-level SY (L/kWh) to solar-driven water harvesting rates in L /day/m². In the Methods section, the use of a 20% PV conversion efficiency to go from solar primary energy to electricity is mentioned to obtain the results in Figure 2b. Was this conversion efficiency also used in the evaluation of the other devices? How do you separate the treatment of input energy of electric-powered systems driven by PV vs solar-thermal systems? Because you are dealing with both solar-PV and solar-thermal systems, when you discuss SY you should state whether this is in kWh electric or kWh primary solar. These points should be addressed in the manuscript.
2. An assumption that is made in the paper is that the daily water harvesting rates (L /day/m²) for devices utilizing sorbents are not impacted or rate-limited by the sorbent regeneration step. In real commercial devices utilizing sorbents, perhaps there is continuous simultaneous sorbent water release and regeneration. Or perhaps if there is not simultaneous water release and regeneration, the quantity of sorbent is vastly "oversized" to allow for unrestricted water release during the coincident times when solar energy is available. However, this is not the case in many of the continuously cycling devices which have been reported in the literature.^{1,2} The authors should explain this assumption and justify why this assumption can be made for the specific devices examined in the manuscript, and how the issue of non-continuous cycling fits into their modeling framework.
3. Line 104 – 106: It is stated that these operational parameters have been achieved experimentally in recent prototypes. However, in the paper cited in reference 14 in the manuscript I do not see a discussion of these specific operational parameters, particularly the energy consumption of recent prototypes.
4. Table 1: Dew Harvesters and Fog Harvesters are listed under passive devices in this table. Please add an explanation of what these devices are and how they differ from each other as there is no accompanying explanation of these devices in the text.
5. Line 152: Table 1 is described as "categorization of AWH devices with low or no power requirements." The row for power requirements for active devices should give power consumption in terms of specific yield (SY) instead of the solar flux.
6. Line 208-209: It should be mentioned that specific yield (SY) is the inverse of a very commonly used

metric in water and desalination systems, specific energy consumption (SEC).

7. Line 223-224: It should be clarified what is meant by active cooler-condensers. Do these devices also use sorbents or do they just rely purely on refrigeration? Active cooler-condenser types of devices don't appear to be addressed in the introduction or included in Table 1 (if they are, it's unclear because the nomenclature is different). A clear explanation in the introduction of the different types of AWH devices (and consistent nomenclature) is needed.

8. Line 228: Please provide a reference for this statement about MOF SY topping out at 1 – 2 L/kWh in most locations.

9. Figure 3d: What are the specific references for the devices using MOFs that are included in this plot? Additionally, the authors should be careful to distinguish between material-level performance and device-level performance as these can differ dramatically.

10. Line 661: It is unclear what is meant by "inside the box". This should be clarified.

11. Line 684: In the paper by Kim et al., the thermal efficiency (and therefore SY), is dependent on heat source temperature, T_H . In the analysis presented showing the water harvesting rates based on Kim et al., it is not stated what heat source temperature is assumed. This information must be included as it impacts the SY.

12. Line 696: It is unclear what is meant by "ventilation energy". Further explanation should be provided.

13. Line 780: If a section to define terms is included, the authors should include and define a more comprehensive list of terms used in the paper.

14. Line 805: It is unclear why some of the results for number of people without SMDW are negative numbers. This should be explained.

15. There are inconsistencies in variable names, labels, legends, superscripts, and subscripts across many of the figures. For example, in Figure 2 AWH output is plotted in L/day/m² but the labels placed on figure 2a are listed in L/d. The manuscript should be checked for consistency of all these elements.

16. The section on Thermal Energy Limits starting on line 639: Please check for consistency among variable names used here and throughout the paper. All variables should also be defined in the text.

17. Extended Data and Supplementary Information section: There is an overall lack of explanation of these tables and figures and a lack of table titles and figure captions. These should be added.

1 Hanikel, N. et al. Rapid Cycling and Exceptional Yield in a Metal-Organic Framework Water Harvester. ACS Central Science 5, 1699-1706, doi:10.1021/acscentsci.9b00745 (2019).

2 Terzis, A. et al. High-Frequency Water Vapor Sorption Cycling Using Fluidization of Metal-Organic Frameworks. Cell Reports Physical Science 1, 100057, doi:https://doi.org/10.1016/j.xcrp.2020.100057 (2020).

Referee #2 (Remarks to the Author):

I was tasked with a somewhat limited review for this manuscript, focused on the use of the ERA5-Land reanalysis and related meteorological calculations. For this reason I do not provide my feedback on the device/materials and water sustainability angles, nor on the originality and significance of the study.

The manuscript assesses the potential of Atmospheric Water Harvesting (AWH) for the technology of solar-driven, continuous-mode AWH (SC-AWH) by estimating the theoretically maximal specific yield (SY) in L/kWh and L/day/m².

The authors extract relevant atmospheric quantities (2m temperature, t_{2m} , 2m dewpoint temperature, d_{2m} and global horizontal irradiance from sunlight, GHI). Key parameters are 2m relative humidity (rH) which is evaluated from d_{2m} and t_{2m} , and GHI. Results take account for the size of population that has no access to safely managed drinking water (SMDW).

As far as I can judge the employed methods are sound, and in my view these would not withhold the acceptance of this manuscript. I do, however, have a number of points I would like the authors to address, as detailed below.

More general comments:

1) I wondered why the authors consider simultaneous thresholds for rH and GHI. I could imagine that SC-AWH works well for more than one quadrant defined by $rH > rH_{ref}$, $GHI > GHI_{ref}$. For instance, higher rH would allow for lower GHI to be working effectively. Could the authors comment on this?

2) To me there seems to be an inconsistent use of the term SY: L/day, L/day/m² or L/kWh. L/day/m² sounds more correct to me than L/day. Do the authors agree on this? Also, for my personal understanding, is it true that L/day/m² is the same unit as mm/day; a quantity that may be less intuitive for this application, but which does allow for a direct scale comparison with precipitation.

3) In their presentation the authors limit themselves to yearly-averaged results. I can imagine that seasonal variations can be quite significant. On the whole, a particular location may look viable, though this could well be only for a part of the year. Could the authors comment on this?

4) I found the supplementary material difficult to read and experienced it as a collection of derivations, tables and plots which could benefit from more coherence. Could the authors comment on my view?

More specific comments:

5) Could the authors confirm what ERA5-Land parameter is used to estimate GHI. Is it 'Surface solar radiation downwards'?

6) Why do the authors limit themselves to the period for 2013-2018. ERA5-Land is available for a longer period and also for most of 2020.

7) Line 631/632: Could the authors explain why ERA5-Land GHI and 2m temperature values are clamped, and indicate where this has an impact?

8) Line 655. In case the unit of rH is in percentage (line 627), strictly speaking it should read $\ln(100/rH_{in})$, correct?

9) Line 681. SY will depend on the value of T_{hot} . For instance, for obvious reasons, $T_{hot} = T_{ambient}$ will result in zero yield. As far I could see the manuscript does not explicitly elaborate on the range of values of this important quantity. Could the authors comment on this?

10) When SC-AWH extracts water from the air, unless fresh ambient air is refreshed adequately, relative humidity will fall, and so will further yield. I presume though that the technology is efficient in refreshing the air. A sentence on this may be useful information for non-expert readers.

Referee #3 (Remarks to the Author):

A. The key results of the paper include datasets to evaluate the feasibility of AWH around the globe, in particular the geospatial tool and analysis/curves for the global potential for AWH. The feasibility of AWH is not only dependent of the data processed but also of aspects such as technological development, governance of water management, and local adoption of the technologies. Despite the additional challenges that these aspects pose, the datasets are useful and are a valuable contribution to water science and to identifying solutions to global water challenges. The point here is that it all depends on the device used. It will be useful to learn what are the recommendations for field experiments, describe what is the path forward, and make reference to how the context, including governance and financing of projects, can impact the assimilation of the technology.

B. This study is original and makes use of Google Earth Engine for a valuable application of data processing about water access and AWH around the world. It also seems to fill in a gap for continuous/active off-grid AWH devices. Although this application is useful, there also seem to be gaps for all other AWH categories presented in Table 1. Explaining which other data and information gaps exist for other AWH devices could help in providing a broader context for the study.

C. For the data analysis, one main observation is the validity of the ERA5 climate data. Although the scientists may trust this dataset, when working on water issues around the world, local validation of data is essential. Although some of the validation may be presented in the appendix, it will be useful if the manuscript main text explains why and how this dataset is superior to others, and why it was selected. Also, any of the results from this study will be impacted by climate change. Showing some sense of the variability around climate as a critical factor, and a preliminary evaluation of the sensitivity of the results to climate variability, will also provide a better context for demonstrating the results strength. The methodology seems solid, although the usability of the results, in particular of Figure 3, could be strengthened with an example of applicability. The reference to India or the SSA is useful, but it doesn't walk the reader through the use of the a, b, c and d graphs. The main question here is how can water managers use this curve to explore the use of these devices as solutions? Give an example for practical use. How can the target curve become real? According to figure d, Only the most optimistic option reaches 1.5 bill/ What is the least optimistic? Is it worth the effort? Explain this in the text.

D. Statistics seem well presented in the appendix. The main missing uncertainty is the climate uncertainty which may provide a wide set of results depending on climate projections. Some evaluation

of this uncertainty will really benefit the paper for credibility.

E. The discussion presents and summarizes well the main results of the paper.

F. The main improvements suggested are:

- including climate change analysis and some sense of how climate can impact the outputs in the presentation of results
- giving an example of the use of Figure 3
- commenting on the other aspects of global water access that can make or not this study applicable

Other specific question for consideration and review in the text include:

In the geography section, water quality for cooking is classified as not important, but that is not correct.

Although high density sites are important, it also masks some of the greatest global water challenges in other regions when combined with economic capacity of countries to solve issues. Please comment about this.

In the presentation of the geospatial tool, there needs to be clarification of why using ERA5 is appropriate. Although this sentence is included 'A longer-term climate record should be used to adjust for decadal climate trends or further correction of interannual variability' the reader is left with a justification of why the dataset was used.

The 'output table' is mentioned, but it is not explained what it contains.

Some of the results for the Andean region don't seem realistic, and the analysis seem to mask the effect of the Andean mountains. There are dry areas that appear wet possibly because of the averaging in the algorithms.

Review figure 2 a, b, and c as the red dots representing density of people without SMDW varies between all images, and what the text indicates is that it should be constant.

What does the 'given' mean in this sentence? 'The results show significant water production potential given small solar collection areas throughout much of the world, particularly in the tropics.'

G. References seem appropriate. I was not familiar with this literature and it seems comprehensive.

H. In the abstract, it will be useful to clarify what is the typical per capita use of water and how many liters per day each person needs. Clarifying the limiting factors will provide context on why AWH is used only for hydration and not for other uses.

When presenting that the study evaluates performance, it may be misleading as we are not evaluating all available devices, but more imagining the targets for devices to have the performance needed. The study is quite hypothetical at this point, and it will take some effort to perform the field studies to validate the information. It is important to make sure the reader understand that from the onset.

When presenting specific yield, it will be useful to provide context on energy consumption for other energy needs for water such as desalination, surface water treatment or groundwater pumping. Solar energy could also be used for such energy needs.

Need to define rH in abstract for reading, and also some definition of what novel sorbents are.

Market serviceability is a sophisticated term that may not resonate with the audience.

When presenting global goals in the last sentence of the abstract, you could be more specific and make reference to the sustainable development goals.

In the intro, SDGs need to be defined. Also, although the comparable table is useful, it leaves questions about how big the devices are, what is their cost, and what are the energy range needs for each type.

Marisa Escobar, PhD
Water Program Director
Stockholm Environment Institute

Author Rebuttals to Initial Comments:

Response to Reviewer #1 Comments

Comments in **bold**, responses in plain text

1. **In the paper it is unclear how the GHI is treated to convert device-level SY (L/kWh) to solar-driven water harvesting rates in L /day/m². In the Methods section, the use of a 20% PV conversion efficiency to go from solar primary energy to electricity is mentioned to obtain the results in Figure 2b. Was this conversion efficiency also used in the evaluation of the other devices? How do you separate the treatment of input energy of electric-powered systems driven by PV vs solar-thermal systems? Because you are dealing with both solar-PV and solar-thermal systems, when you discuss SY you should state whether this is in kWh electric or kWh primary solar. These points should be addressed in the manuscript.**

In the paper, the energy units in L/kWh are incident solar energy directly from GHI. The reported values have various assumptions based on their source. The Thermal Limits (Kim 2020), Characteristic Curves, and experimental results reported by SMAG (Zhao 2019) & MOFs were directly (100%) conversion from sunlight to work. For ZMW SOURCE, the table provided by the manufacturer accounts for system losses, so the table values were directly converted in our model. Bagheri's results and the Cooler-Condenser limits from Peeters were applied a PV conversion efficiency between 20 - 34%.

We added clarity to the paper in the main and methods section to address this ambiguity.

Lines 119-125, 299, 638-647, 753-759, 815-816

2. **An assumption that is made in the paper is that the daily water harvesting rates (L /day/m²) for devices utilizing sorbents are not impacted or rate-limited by the sorbent regeneration step. In real commercial devices utilizing sorbents, perhaps there is continuous simultaneous sorbent water release and regeneration. Or perhaps if there is not simultaneous water release and regeneration, the quantity of sorbent is vastly "oversized" to allow for unrestricted water release during the coincident times when solar energy is available. However, this is not the case in many of the continuously cycling devices which have been reported in the literature.^{1,2} The authors should explain this assumption and justify why this assumption can be made for the specific devices examined in the manuscript, and how the issue of non-continuous cycling fits into their modeling framework.**

Thanks for flagging, this is important. There are assumptions to our modeling framework which needs more clarity in the text of the paper, but this is also a complex discussion which cannot yet be resolved in a single study. We've added some to the manuscript to clarify, and discuss more here.

Simply put, our model does not account for the mechanics of sorbent cycling. Each timestep of the time series is treated independently -- in the case of ERA5 (our primary climate time series

dataset), this is hourly. Therefore, any asynchrony of solar energy input to sorbent cycling dynamics beyond 60 minutes is not explicitly modelable using our tools as currently published. For each (hourly) climate timestep, at each geographic pixel, a single (3-tuple) value of GHI, rH, and temp (representing the average of each over the hour period) is converted to an AWH (device) output in liters (or L/h/m²). This makes the implicit assumption of near-instantaneous continuous sorbent cycling, including water release and regeneration, or, perhaps, a cycle with a single hour duration.

As you rightly point out, there are many classes and species of sorbents (MOFs, SMAGs, silica, liquid desiccants, etc), sorbent configurations (wheels, fluidized beds <Terzis 2020>, matrices <Yilmaz 2020>), and device architectures (recirculated and multi-staged air streams <LaPotin 2020>, etc), all of which have some kinetic and/or cycle rate limits. “Continuous cycling” may be itself somewhat of a misnomer, since there are likely no sorbent processes which are truly instantaneous, though some are approaching this <Qi 2019>. This is a limitation of our framework -- each device will have to be modeled with this in mind, and device designers need to populate an Output Table with water output values that best-approximate the output within an hour given the climate inputs, or an output rate which could be achieved if those climate conditions were present.

If a device is “near-continuous”, and has a cycle duration of a several hours, it may regenerate the sorbent (collect water) during higher rH periods of the morning hours and use the high solar energy of midday and afternoon to finalize water release. Or some mix thereof. Some changes to our model could be made in Earth Engine to use n previous timesteps as input to the current timestep’s water output value, but we leave this up to the user to do, and our code is open to the community.

All that said, through our work developing a real sorbent-based device prototype which successfully operated in continuous mode, and appreciating the transformative value in the market of a device which can be modular and sized to a single household or individual, we are “picking the horse” of continuous mode and want to push the AWH community in this direction with this framework and study as presented. We are expressly not including diurnal or batch-mode devices in our assessment -- those which require regeneration entirely at night and release during the day when sunlight is available. This necessarily requires large sorbent beds, higher mass, and ultimately higher cost. New sorbents and configurations in continuous mode show promise and (we think) should be aggressively pursued and scaled to best meet the global challenge.

Finally, beyond sorbent-based architectures, our model as presented provides compatibility with cooler-condenser devices. Though they are less likely to “beat” solar-thermal sorbent devices given the low conversion efficiency from solar to electricity using PV (as discussed in other responses here), they are indeed continuous and can be run through our geo-processing without much customization. Our tools and global assessment were built assuming continuous mode, but generalized across device categories. We think it strikes a fair balance between

abstraction (for a broad global assessment) and specificity (for inventors of real devices to do design trade-offs).

Lines 649-653

- 3. Line 104 – 106: It is stated that these operational parameters have been achieved experimentally in recent prototypes. However, in the paper cited in reference 14 in the manuscript I do not see a discussion of these specific operational parameters, particularly the energy consumption of recent prototypes.**

Reference 14 of the original manuscript was Hanikel 2020, a review of MOF harvesters and contains recent experimental results. There are many experiments which have achieved or are approaching these target values, and instead of citing many we cited the review. However, this is confusing and not precise, and it's been removed.

Reference 15 (Zhao 2019 on SMAG material) was, however, an experimental study which did achieve these results, so we left it in the revised manuscript. Zhao writes: "It only takes ≈ 100 , 120, and 180 min for the SMAGs to be sufficiently hydrated by capturing water from air at RH of 90%, 60%, and 30%, respectively"; "Subsequently, the hydrated SMAGs were retrieved and exposed to the sunlight ($\approx 0.7 \text{ kW m}^{-2}$) from 10:00 a.m. to 2:00 p.m. (Figure 5e)". These results were cross-referenced with Peeters 2020 who converted values to SY in L/kWh, which far overshoots the target SY values required for 1 bil ppl without SMDW we report. It seems thus likely that a device could be engineered to achieve this in field conditions.

We've also added the landmark Kim 2017 study which also achieves the target yields.

Line 88.

- 4. Table 1: Dew Harvesters and Fog Harvesters are listed under passive devices in this table. Please add an explanation of what these devices are and how they differ from each other as there is no accompanying explanation of these devices in the text.**

We revised / added some language to the text and to the table to clarify. However, we want to avoid writing a review paper of AWH devices, since there are several in print. Our point in the introduction is to reframe AWH categorization to highlight gaps in literature, on which category our study focuses, and, most importantly, why a solar-driven continuous device shows most promise for impact at the household level.

Lines 110-112.

- 5. Line 152: Table 1 is described as "categorization of AWH devices with low or no power requirements." The row for power requirements for active devices should give power consumption in terms of specific yield (SY) instead of the solar flux.**

Good point, we've modified.

- 6. Line 208-209: It should be mentioned that specific yield (SY) is the inverse of a very commonly used metric in water and desalination systems, specific energy consumption (SEC).**

Good idea. We've added.

Line 202-203

- 7. Line 223-224: It should be clarified what is meant by active cooler-condensers. Do these devices also use sorbents or do they just rely purely on refrigeration? Active cooler-condenser types of devices don't appear to be addressed in the introduction or included in Table 1 (if they are, it's unclear because the nomenclature is different). A clear explanation in the introduction of the different types of AWH devices (and consistent nomenclature) is needed.**

We've added additional language to describe cooler-condensers. However, we purposely created a somewhat odd categorization of AWH to steer our reader (and the research community) towards SC-AWH. We are not writing a review of AWH types, but rather reviewing enough to get the reader to understand how SC-AWH fits. We think solar thermal sorbent-based devices will be the winner here, but we never say this directly. Technically, the race is still on.

Line 119-122.

- 8. Line 228: Please provide a reference for this statement about MOF SY topping out at 1 – 2 L/kWh in most locations.**

This was poorly worded, now updated. The citation is from Peeters 2020 (now added), they top out around 1 L/kWh, and we wrote "in most locations" based on our own geographic analysis.

Line 222

- 9. Figure 3d: What are the specific references for the devices using MOFs that are included in this plot? Additionally, the authors should be careful to distinguish between material-level performance and device-level performance as these can differ dramatically.**

The MOF material yields come from Hanikel 2019 and Wang 2017 converted to SY by Peeters 2020. We've added references. These are results from experiments of MOFs in laboratory conditions, not devices.

The framework we've established to relate abstract SY values to impact gives "apples-and-oranges" comparisons between materials, devices, or anything in between. This allows researchers of each to coordinate towards devices which have the highest market size and human impact.

However, the referee's point is well-taken, and apparently this was not communicated clearly enough in the submitted manuscript. We've added language to the text to highlight to the reader that we're comparing material and device performances side-by-side.

Line 298.

10. Line 661: It is unclear what is meant by "inside the box". This should be clarified.
Yes, this part of the method section was unclear and has been rewritten.

Lines 698-751.

11. Line 684: In the paper by Kim et al., the thermal efficiency (and therefore SY), is dependent on heat source temperature, TH. In the analysis presented showing the water harvesting rates based on Kim et al., it is not stated what heat source temperature is assumed. This information must be included as it impacts the SY.
New language in the methods addresses this point.

Line 746.

12. Line 696: It is unclear what is meant by "ventilation energy". Further explanation should be provided.
Ventilation energy is an electric-powered fan to circulate air. We removed this since it's unclear and unnecessary.

13. Line 780: If a section to define terms is included, the authors should include and define a more comprehensive list of terms used in the paper.
We've removed this line. Terms are defined as they arise in the paper.

14. Line 805: It is unclear why some of the results for number of people without SMDW are negative numbers. This should be explained.
This was a table showing the delta in results between input datasets. It has been removed in favor of the charts only.

15. There are inconsistencies in variable names, labels, legends, superscripts, and subscripts across many of the figures. For example, in Figure 2 AWH output is plotted in L/day/m² but the labels placed on figure 2a are listed in L/d. The manuscript should be checked for consistency of all these elements.
We've updated the manuscript to be consistent with these units and symbols.

16. The section on Thermal Energy Limits starting on line 639: Please check for consistency among variable names used here and throughout the paper. All variables should also be defined in the text.
Yes; the reviewer points out a weakness in our presentation of this method section, which we saw as well. We rewrote this section and expect it to be much clearer now. Variable names have

been harmonized and each equation is now followed by listing describing the meaning of any new variable or symbol (we will happily follow Nature's style guidance in how these are laid out, formatted or shortened).

Lines 698-751.

17. Extended Data and Supplementary Information section: There is an overall lack of explanation of these tables and figures and a lack of table titles and figure captures. These should be added.

We updated and reformatted the ED.

Response to Reviewer #2 Comments

Comments in bold, responses in plain text

- 1. I wondered why the authors consider simultaneous thresholds for rH and GHI. I could imagine that SC-AWH works well for more than one quadrant defined by $rH > rH_{ref}$, $GHI > GHI_{ref}$. For instance, higher rH would allow for lower GHI to be working effectively. Could the authors comment on this?**

While Fig4c-d gives performance projections for certain device characteristics, Fig4a-b asks and answers the question: “for how many hours do GHI and rH coincide above thresholds?”. This provides data to contradict a prevalent assumption in prior literature (citations 7-11) that humidity and sunlight would not sufficiently coincide for viability of continuous mode AWH.

The referee is correct that these binary thresholds do not represent how a typical device would operate. Instead, they are the lowest-parameter construct we could come up with to inform device design. The results make visible the critical need to (i) design AWH for operation down to modest humidities around 30%rH and (ii) highlight the attention that needs to be given to device capability under 400-600 W/m² of GHI. The ED figure shows more parameter range than Fig4a-b is able to, given space limitations.

- 2. To me there seems to be an inconsistent use of the term SY: L/day, L/day/m² or L/kWh. L/day/m² sounds more correct to me than L/day. Do the authors agree on this? Also, for my personal understanding, is it true that L/day/m² is the same unit as mm/day; a quantity that may be less intuitive for this application, but which does allow for a direct scale comparison with precipitation.**

We've updated the manuscript to be consistent with these units and symbols. We also added some description of how AWH-Geo can be used to model AWH devices in L/d rather than abstract areal units.

- 3. In their presentation the authors limit themselves to yearly-averaged results. I can imagine that seasonal variations can be quite significant. On the whole, a particular location may look viable, though this could well be only for a part of the year. Could the authors comment on this?**

We've added significantly to the paper on variability and seasonality of AWH output.

Lines 322-332, 865-882, Extended Data pages 3-8.

- 4. I found the supplementary material difficult to read and experienced it as a collection of derivations, tables and plots which could benefit from more coherence. Could the authors comment on myview?**

We updated and reformatted the Extended Data.

- 5) Could the authors confirm what ERA5-Land parameter is used to estimate GHI. Is it 'Surface solar radiation downwards'?**

Yes, correct 'Surface solar radiation downwards'. We originally reported as 'Surface net solar radiation', which was incorrect.

Line 682-683.

6. Why do the authors limit themselves to the period for 2013-2018. ERA5-Land is available for a longer period and also for most of 2020.

Originally, we limited our time period to 5 years to allow researchers to run the tool within reasonable computation limits. For this revision, we extended the period of our analysis to 10 years (2010-2020), and added functionality to the tool to allow users to run any yearly time window of their choosing.

7. Line 631/632: Could the authors explain why ERA5-Land GHI and 2m temperature values are clamped, and indicate where this has an impact?

We clamped these values out of simplicity and since our focus was in tropical regions. However, since we're attempting a global analysis, we decided to remove the clamps, bring the output table interval values for temperature down to 0 deg C, and assume an output rate of 0 L/h at 0 deg C and below. At freezing temperatures, though water vapor could technically be extracted, the liquid water would freeze in the output chamber of the device. We also extended the GHI intervals from 1200 W/m² to 1300 W/m² which is nearly the solar constant, thus obviating the need for a clamp since no lookup value will reach above this on the surface.

8. Line 655. In case the unit of rH is in percentage (line 627), strictly speaking it should read $\ln(100/rH_in)$, correct?

The reviewers suggestion is a possibility (with a small modification). We leave it to Nature's editorial preferences to pick between the two valid forms:

$\ln(1/rH_in)$ or $\ln(100\%/rH_in)$

Since a relative humidity of e.g. 67% is equal to 0.67 (dimensionless), we prefer $\ln(1/rH_in)$, but are comfortable with whatever form will be best received by readers.

9. Line 681. SY will depend on the value of T_hot. For instance, for obvious reasons, T_hot = T_ambient will result in zero yield. As far I could see the manuscript does not explicitly elaborate on the range of values of this important quantity. Could the authors comment on this?

As added to our methods section:

While this can be run for any choice of parameter T_{hot} , we present figures here for $T_{hot}=100$ °C, a temperature still achievable in low-cost (non-vacuum) practical devices without tracking or sunlight concentration. Higher driving temperatures increase the upper bound for water output.

10. When SC-AWH extracts water from the air, unless fresh ambient air is refreshed adequately, relative humidity will fall, and so will further yield. I presume though that the technology is efficient in refreshing the air. A sentence on this may be useful information for non-expert readers.

Good idea, we've added some language in the Methods. However, detailed considerations of specific design or device architecture such as air circulation methods are, we believe, beyond our scope.

Lines 750-751.

Response to Reviewer #3 Comments

Comments in bold, responses in plain text

- 1. Although this application is useful, there also seem to be gaps for all other AWH categories presented in Table 1. Explaining which other data and information gaps exist for other AWH devices could help in providing a broader context for the study.**

This is a good point, but with limited word counts and our focus on SC-AWH and belief of its singular promise for AWH, we decided not to add reviews of data gaps for other AWH categories.

- 2. For the data analysis, one main observation is the validity of the ERA5 climate data. Although the scientists may trust this dataset, when working on water issues around the world, local validation of data is essential. Although some of the validation may be presented in the appendix, it will be useful if the manuscript main text explains why and how this dataset is superior to others, and why it was selected. Also, any of the results from this study will be impacted by climate change. Showing some sense of the variability around climate as a critical factor, and a preliminary evaluation of the sensitivity of the results to climate variability, will also provide a better context for demonstrating the results strength. The methodology seems solid, although the usability of the results, in particular of Figure 3, could be strengthened with an example of applicability. The reference to India or the SSA is useful, but it doesn't walk the reader through the use of the a, b, c and d graphs. The main question here is how can water managers use this curve to explore the use of these devices as solutions? Give an example for practical use. How can the target curve become real? According to figure d, Only the most optimistic option reaches 1.5 bill/ What is the least optimistic? Is it worth the effort? Explain this in the text.**

Thank you, the reviewer raises good points. We've added some language into the main text of the high spatial and temporal resolution, global coverage, and seamless access to ERA5, which makes it the best selection for our work. We've also added some analysis on decadal trends and a lot on seasonality. We've expanded a bit on our observation of global trends in AWH output potential and how design tradeoffs (e.g. SY profile) can affect suitability in different regions. However, we think providing more specific point-by-point examples of the use of our main figures is out of scope of our work. There are tight word counts and a lot of high-level material which needs to be covered and is not covered in other literature specific to SC-AWH. How exactly our results are used by the wide range of materials and device researchers, climate and environmental scientists, and water managers, and how and whether SC-AWH is worth the effort to develop, are questions that we don't feel fit to answer directly in the text.

- 3. Statistics seem well presented in the appendix. The main missing uncertainty is the climate uncertainty which may provide a wide set of results depending on**

climate projections. Some evaluation of this uncertainty will really benefit the paper for credibility.

We've added some analysis on decadal trends in ED page 9.

4. ...including climate change analysis and some sense of how climate can impact the outputs in the presentation of results

This is a good point, but our study is not a climate study per se, and we think a discussion of how climate change will affect SC-AWH output is too complex and out of scope to include in the text. A full understanding of the global AWH resource is still in its nascency, and SC-AWH in particular. This study hopes to begin this. Many other uncertainties in our analysis, such as SY profiles, device losses and real-world conditions, and device durability and longevity, are primary before an assessment of climate change within reasonable ranges of uncertainty should be attempted.

5. ...commenting on the other aspects of global water access that can make or not this study applicable

We added several points regarding the use of alternate sources, rainwater and storage and contamination, user behavior, the benefits to environmental waste from replacing plastic bottled water, and the overall limits to technology development in the overall global drinking water challenge.

Lines 322-332, 348-354, 102-105.

6. In the geography section, water quality for cooking is classified as not important, but that is not correct.

Fair point. We said "less important", in reflection of the stratification of water needs. We have removed that statement as the core of the argument is the water is sufficient quantity and quality for drinking purposes alone.

Line 147-149.

7. Although high density sites are important, it also masks some of the greatest global water challenges in other regions when combined with economic capacity of countries to solve issues. Please comment about this.

The central thesis of this paper is that AWHs have the potential to drastically reduce the population of people living without SMDW as the geographies with appropriate climatic conditions are largely coincident with the geographies with the most need. Our analysis focused on the areas with the highest populations in need of SMDW to demonstrate the viability of AWHs. Some of our commentary was around areas with high need, but variable AWH potential due to the steep gradient of climatic conditions (Sahel of Africa).

Most countries with low access to SMDW fall within the high output AWH regions. However, we do not intend to claim AWHs can solve the drinking water need globally; areas such as the

northern Sahel will need alternative technologies as the predicted AWH output is insufficient to meet drinking water needs.

How the water infrastructure (whether AWH or traditional) will be financed, particularly by countries with low economic capacity is a critical topic, though outside the scope of what we were able to include in this paper.

- 8. In the presentation of the geospatial tool, there needs to be clarification of why using ERA5 is appropriate. Although this sentence is included ‘A longer-term climate record should be used to adjust for decadal climate trends or further correction of interannual variability’ the reader is left with a justification of why the dataset was used.**

We’ve reworded, added some detail of why ERA5 is appropriate, and added some analysis on decadal trends.

Lines 172-176, ED page 9.

- 9. The ‘output table’ is mentioned, but it is not explained what it contains.**

The output table is described in the main text and in more detail in the methods. We’ve added a few phrases for further description.

Lines 182-187, 662-669.

- 10. Some of the results for the Andean region don’t seem realistic, and the analysis seem to mask the effect of the Andean mountains. There are dry areas that appear wet possibly because of the averaging in the algorithms.**

Good eye. This may have been an artifact of our data processing bug or the effect of clamping temperature values below 10 deg C (from reviewer #2, comment 7). We’ve updated both (now clamp values below 0 deg, which will certainly affect mountainous regions), and the results are more realistic.

- 11. Review figure 2 a, b, and c as the red dots representing density of people without SMDW varies between all images, and what the text indicates is that it should be constant.**

The dots are spaced arbitrarily within each survey region since that is their finest spatial resolution. There are also some difficulty in scaling the dots size when copying from PNG to Word to PDF which can be fixed in the final proof.

- 12. What does the ‘given’ mean in this sentence? ‘The results show significant water production potential given small solar collection areas throughout much of the world, particularly in the tropics.’**

Changed to “using”.

Lines 214-215.

13. In the abstract, it will be useful to clarify what is the typical per capita use of water and how many liters per day each person needs. Clarifying the limiting factors will provide context on why AWH is used only for hydration and not for other uses.

Good idea. We've now added our drinking water requirement target of 5 L/d/person to the abstract, citing WHO and other studies.

Line 80.

14. When presenting that the study evaluates performance, it may be misleading as we are not evaluating all available devices, but more imagining the targets for devices to have the performance needed. The study is quite hypothetical at this point, and it will take some effort to perform the field studies to validate the information. It is important to make sure the reader understand that from the onset.

We have responded to this in detail above, in our response to the Editor's highlight. We have added language to the manuscript and some justification for the hypothetical nature of our work.

We also added a plug for field validation of real devices.

Lines 319-320.

15. When presenting specific yield, it will be useful to provide context on energy consumption for other energy needs for water such as desalination, surface water treatment or groundwater pumping. Solar energy could also be used for such energy needs.

We think this is covered adequately in many other studies and is out of scope for our work. However, to position SC-AWH as a technology suitable for household-scale drinking water output, we added a SY value of desalination as comparison (250 L/kWh). This aligns well with our overall argument that AWH should rely on its unique advantages and be considered an off-grid, decentralized option for increasing drinking water access prior to full infrastructure buildout, rather than going head-to-head with traditional source.

Lines 137-138.

16. Need to define rH in abstract for reading, and also some definition of what novel sorbents are.

We've added the rH term definition in the abstract and reworded the point about sorbents.

Lines 73, 87-88

17. Market serviceability is a sophisticated term that may not resonate with the audience.

Agreed, this is vague. We have removed this term in favor of new wording in the abstract.

Lines 88-90

18. When presenting global goals in the last sentence of the abstract, you could be more specific and make reference to the sustainable development goals.

Done.

Line 91.

19. In the intro, SDGs need to be defined.

Done.

Line 95.

20. Also, although the comparable table is useful, it leaves questions about how big the devices are, what is their cost, and what are the energy range needs for each type.

The table is meant to provide a very conceptual overview of why SC-AWH is most appropriate for small, inexpensive devices at the household or individual scale. We don't think more detail is needed, and may serve to obscure our message.

Reviewer Reports on the First Revision:

Referee #1 (Remarks to the Author):

Review of "Global Potential for Harvesting Drinking Water from Air using Solar Energy" (manuscript number 2020-12-21648B) by Jackson Lord et al.

This manuscript describes a new model which uses specific energy consumption performance data from atmospheric water harvesting (AWH) devices combined with global climate data and solar energy availability (global horizontal irradiance, GHI) to determine potential global water harvesting rates using solar energy. The model also uses data about population without safely managed drinking water (SMDW) to determine the global population whose drinking water source could be met using AWH. This work is interesting because it is the most comprehensive that I am aware of which provides a continuous global assessment of the potential of AWH based on global climate data and device performance. The work uses specific energy consumption from real commercial devices, devices in literature, and theoretical thermodynamic minimum specific energy consumption. It creates a framework which can be used to assess suitability and potential of AWH for different regions and comparison of devices.

I have reviewed the author's responses to my original comments, and I am satisfied with the improvements they have made to the manuscript. They have improved the clarify and explanation of the Methods and Extended Data. The discussion and clarification regarding the conversion from primary energy to electricity via photovoltaic is now present.

My final minor comments before the manuscript is fit for publication are:

- 1) I am glad to see the clarification between primary energy and PV electrical energy. However, it is not clear to me why the authors decided to use the typical 20% PV conversion efficiency for the results from Bagheri, but they use thermodynamic maximum Shockley-Queisser conversion efficiency of 33.77% for the results from Peters et al. In the case of Peters et al., this was justified by representing an upper bound on electricity-dependent devices (line 758). In that case, why was a 20% conversion efficiency used for the results from Bagheri (line 816)? A consistent PV conversion efficiency is needed to compare the different PV-driven devices to each other.
- 2) A very minor comment: Figure 1, 3, and 4 are a bit messy with a mix of superscripts and "[^]" symbols, a mix of "[]" and "()" symbols and inconsistent font sizes. There is room for aesthetic improvement to these figures.

Referee #2 (Remarks to the Author):

I would like to thank the authors for their resubmission. It looks to me like they went 'at full length' to incorporate the reviewers feedback, including the points that I had raised. I am happy with their revisions.

I am particularly pleased to see:

- 1) an analysis and discussion on seasonal effects
- 2) extension of the period from 5 to 10 years. I think the period they choose (2010-2020) is just right: sufficiently recent, long enough to have reliable statistics but not too long for climate change effects to bias the validity for today's applications. In that respect I was also pleased to see the extended Figure 9 that shows a shift between the 2010s and 2000s.

I only have two very minor points:

1) line 173: "ERA5-Land, a climate model". Suggestion: replace by: "ERA5-Land, a climate reanalysis" ERA5-Land is a dynamical downscaling of the ERA5 climate reanalysis. As such ERA5-Land is much more than a climate model or simulation; it is actually based on large amounts of observations and is therefore able to represent the actual historical synoptic situation. I think that realization will strengthen the case for the credibility of the results in this manuscript.

2) mainly just for clarity: strictly speaking 2010-2020 could be interpreted as a 11 year period (i.e., including the year 2020), which is not the case.

It would help if the authors could state this more clearly.

Referee #3 (Remarks to the Author):

I reviewed the paper and I believe my comments were properly addressed.

Author Rebuttals to First Revision:

Response to Reviewer #1 Comments

Comments in bold, responses in plain text

1. I am glad to see the clarification between primary energy and PV electrical energy. **However, it is not clear to me why the authors decided to use the typical 20% PV conversion efficiency for the results from Bagheri, but they use thermodynamic maximum Shockley-Queisser conversion efficiency of 33.77% for the results from Peters et al. In the case of Peters et al., this was justified by representing an upper bound on electricity-dependent devices (line 758). In that case, why was a 20% conversion efficiency used for the results from Bagheri (line 816)? A consistent PV conversion efficiency is needed to compare the different PV-driven devices to each other.**

Thanks for raising. We used two PV conversion values since the CoolerCondenser scenario from Peeters is an upper limit of possibility (as identified by reviewer), and we figured we should also use an upper limit conversion efficiency of PV modules -- perhaps in the future these advanced modules will come down in cost, we reasoned. With Bagheri, on the other hand, we're showing what's possible today from commercially-available AWH devices, so we used (an approximate) PV efficiency of today's commercially-available modules of 20% (from NREL).

However, since this confused Reviewer #1, we can expect it will confuse others. Moreover, it makes more sense to isolate our analysis to AWH potential rather than conflating it with PV technology development. Therefore, we removed the Shockley-Queisser limit from our analysis in favor of the standard 20% PV conversion for both the Peeters- and Bagheri-derived scenarios throughout the work. Figures and text are now updated in this revision to reflect this.

- 2. A very minor comment: Figure 1, 3, and 4 are a bit messy with a mix of superscripts and “^” symbols, a mix of “[]” and “()” symbols and inconsistent font sizes. There is room for aesthetic improvement to these figures.**

Indeed they were. We're now armed with Adobe Illustrator and have updated all the figures for consistency!

Response to Reviewer #2 Comments
Comments in bold, responses in plain text

- 1. line 173: "ERA5-Land, a climate model". Suggestion: replace by: "ERA5-Land, a climate reanalysis" ERA5-Land is a dynamical downscaling of the ERA5 climate reanalysis. As such ERA5-Land is much more than a climate model or simulation; it is actually based on large amounts of observations and is therefore able to represent the actual historical synoptic situation. I think that realization will strengthen the case for the credibility of the results in this manuscript.**

Thank you, this is a good suggestion. Defending the choice of ERA5-Land was also raised previously by Reviewer #3. To buttress for readers unfamiliar with reanalysis products, we went a step further and now write:

"AWH-Geo uses the climate reanalysis ERA5-Land over the ten year period 2010-2019 (inclusive). ERA5-Land was chosen for its fine resolution (9 km at hourly intervals), global coverage, and ability to represent historical synoptic conditions."

- 2. mainly just for clarity: strictly speaking 2010-2020 could be interpreted as a 11 year period (i.e., including the year 2020), which is not the case. It would help if the authors could state this more clearly.**

Good point, that's unclear. We now write: "...the ten year period 2010-2019 (inclusive)".

Response to Reviewer #3 Comments
Comments in bold, responses in plain text

1. **I reviewed the paper and I believe my comments were properly addressed.**
Thank you for your helpful previous review, the manuscript is now much stronger.